# An Update of Epigenetic Drugs for the Treatment of Cancers and Brain Diseases: A Comprehensive Review

**DOI:** 10.3390/genes14040873

**Published:** 2023-04-06

**Authors:** Zahra Sahafnejad, Shahin Ramazi, Abdollah Allahverdi

**Affiliations:** Department of Biophysics, Faculty of Biological Sciences, Tarbiat Modares University, Jalal Ale Ahmad Highway, Tehran P.O. Box 14115-111, Iran

**Keywords:** epigenetic disorders, epigenetic therapy, DNA methylation, DNA methyltransferases, histone modifications, personalized medicine

## Abstract

Epigenetics has long been recognized as a significant field in biology and is defined as the investigation of any alteration in gene expression patterns that is not attributed to changes in the DNA sequences. Epigenetic marks, including histone modifications, non-coding RNAs, and DNA methylation, play crucial roles in gene regulation. Numerous studies in humans have been carried out on single-nucleotide resolution of DNA methylation, the CpG island, new histone modifications, and genome-wide nucleosome positioning. These studies indicate that epigenetic mutations and aberrant placement of these epigenetic marks play a critical role in causing the disease. Consequently, significant development has occurred in biomedical research in identifying epigenetic mechanisms, their interactions, and changes in health and disease conditions. The purpose of this review article is to provide comprehensive information about the different types of diseases caused by alterations in epigenetic factors such as DNA methylation and histone acetylation or methylation. Recent studies reported that epigenetics could influence the evolution of human cancer via aberrant methylation of gene promoter regions, which is associated with reduced gene function. Furthermore, DNA methyltransferases (DNMTs) in the DNA methylation process as well as histone acetyltransferases (HATs)/histone deacetylases (HDACs) and histone methyltransferases (HMTs)/demethylases (HDMs) in histone modifications play important roles both in the catalysis and inhibition of target gene transcription and in many other DNA processes such as repair, replication, and recombination. Dysfunction in these enzymes leads to epigenetic disorders and, as a result, various diseases such as cancers and brain diseases. Consequently, the knowledge of how to modify aberrant DNA methylation as well as aberrant histone acetylation or methylation via inhibitors by using epigenetic drugs can be a suitable therapeutic approach for a number of diseases. Using the synergistic effects of DNA methylation and histone modification inhibitors, it is hoped that many epigenetic defects will be treated in the future. Numerous studies have demonstrated a link between epigenetic marks and their effects on brain and cancer diseases. Designing appropriate drugs could provide novel strategies for the management of these diseases in the near future.

## 1. Introduction

### 1.1. Epigenetics Overview

“Epigenetics” was initially described by British developmental scholar Conrad Waddington in the 1940s as “a new field of research in biology to study the relationship between genes and their products, such as RNA and protein” [1]. He discovered that variations in ambient heat or chemical factors could cause different thorax and wing structures in the larvae of fruit flies with the exact same history that is hereditary [1]. The phrase “epigenetics” initially encompassed the system through which a fertilized zygote grows into a developed and complex living organism. However, confident results revealed that cells with similar DNA could experience differential regulation of gene function. To achieve this purpose, Waddington’s insight assisted in showing an important biochemical pathway in which genetic characteristics are correlated not just with improvements within the nucleotide sequence but also with chemical alterations of DNA or proteins with which DNA interacts. The development of DNA methylation in bacterial genomes ended up being followed by Waddington’s findings. In 1975, researchers discovered the possibility of inheriting epigenetic patterns into divided daughter cells where gene expression is highly regulated [2]. Epigenetics is categorized into three main sub-classes: (i) chemical modification in DNA (for example, DNA methylation); (ii) post-translational modifications (PTMs) in histone tails; and (iii) changes in regulation of gene expression by non-coding RNAs “ncRNAs” e.g., microRNAs (miRs), P-Element Induced Wimpy test PIWI-interacting RNAs, endogenous short interfering RNAs, and long non-coding RNAs (lncRNAs) [3].

### 1.2. Non-Coding RNAs

Nc-RNAs do not translate into proteins and, during splicing and processing, are divided into housekeeping ncRNAs and regulatory RNAs that are non-coding. However, regulatory RNAs that are non-coding RNAs based on size are categorized as short-chain ncRNAs (such as siRNAs, miRNAs, and piRNAs) and lncRNAs. ncRNAs play a vital role in the regulation of epigenetic modification and lead to the control of cell differentiation via regulating expression at the level of the gene rather than at the level of the chromosome [4]. These three classes of epigenetics are believed to be epigenetic ‘marks’ of an individual as well. They are recognized as the epigenome. In addition, they affect the regulation of gene expression, which can play significant roles in biological processes like DNA repair, DNA replication, growth, and proliferation. Thus, if a mutation or deletion occurs within these “marks”, it can cause a variety of human diseases at the level of transcriptional regulation [5]. Figure 1 depicts the use of ncRNAs as an epigenetic tool in mRNA processing.

Similar to DNA and histone modifications as regular epigenetic modifiers, various chemical modifications on RNAs have also been discovered and named “RNA epigenetics” such as N6-methyladenosine (m6A), N1-methyladenosine (m1A), 5-methylcytidine (m5C), and 7-methylguanosine (m7G) [6,7]. 

#### 1.2.1. N6-Methyladenosine 

The most common internal RNA modification, m6A, typically occurs in close proximity to stop codons and the 3′ untranslated region within an RRACH or DRACH sequence motif (R = A or G; H = A, C, or U) [8].

Functional studies have demonstrated that m6A is involved in nearly every aspect of RNA metabolism and function, such as secondary structure formation, alternative splicing, mRNA translation, and subcellular localization.

The levels of cellular m6A are carefully controlled by many “writers”, “erasers”, and “readers”, indicating the prospect that m6A may play significant roles in cell signaling networks. The activity of m6A is catalyzed by the m6A Methyl Transferase Complex (MTC), which is composed of two “writer” proteins, METhylTransferase-3 and 14 (METTL3) and METTL14 [9]. 

Making a complex between METTL3-METTL13 is required for m6A enzyme formation and activation, which catalyzes a wide range of RNA molecules [10]. m6A is an epigenetic mark that, like DNA and histone methylation, must be interpreted by m6A “reader” proteins to produce a functional signal. The majority of m6A readers belong to the YT521-B homology (YTH) domain-containing protein family, which directly binds to m6A modified RNA bases. While YTHDF2 may have an impact on RNA stability, YTHDF1 regulates the effectiveness of translation [11]. Altogether, m6A is tightly regulated by a variety of “writers”, “erasers”, and “readers” and is essential for the metabolism of RNA, particularly for the processing of miRNAs and the actions of lncRNAs.

#### 1.2.2. m6A Regulates miRNA Processing

miRNAs are highly conserved short RNAs that are involved in a variety of pathogenic events, such as developmental disorders and the spread of cancer [12]. In earlier research, by interacting with the miRNA processor protein DGCR8, m6A was found to be involved in the processing of miRNAs [13]. Notably, cancer cells showed a similar process. In a bladder cancer model, a recent study evaluated the relationship between METTL3 and DGCR8. Results revealed that METTL3 increased the binding of DGCR8 to pri-miRNA-221/222 through its m6A activity, which accelerated the maturation of miR-221/222’s and decreased the tumor suppressor PTEN, the target of miR-221. This encouraged the proliferation of bladder cancer cells both in vitro and in vivo. Additionally, METTL3 promoted the maturity of miR-1246, which in turn decreased the expression of the tumor suppressor gene SPRED2 and increased the ability of colorectal cancer cells to metastasize [14,15]. In conclusion, miRNAs regulated by m6A and their downstream target genes are mostly involved in the pathways and lead to cell proliferation and metastasis, suggesting that miRNAs may be a crucial link in the chain that m6A uses to control the spread of cancer.

#### 1.2.3. 7-Methylguanosine (m7G)

At the 5′ cap of eukaryotic mRNA, m7G is a crucial modifier that controls mRNA translation, sub-location, and splicing. Furthermore, human cytoplasmic tRNA and 18S-rRNA are both regulated by m7G, which is stabilized by the WBSCR22 and METTL1-WDR4, m7G “writer” complexes [15].

Barbieri et al. demonstrated that a class of tumor suppressor miRNAs, such as let-7e, contain m7G modification sites using borhydride reduction sequencing (BoRed-seq) and mass spectrometry technology [16]. The results of a subsequent assay demonstrated that m7G methyltransferase METTL1 is able to process and mature these miRNAs by binding directly to their precursors. 

#### 1.2.4. 2-O-Methylation (2-O-Me)

The RNase III enzymes, Drosha and Dicer, successively cleave the main transcript of canonical miRNAs to produce a 5′ monophosphate end that is crucial for later miRNA activities [17].

For precise and efficient miRNA maturation, Dicer must specifically recognize the 5′ monophosphate of pre-miRNAs. BCDIN3D is a potential Sadenosyl Methionine (SAM) binding motif-containing human ortholog of the Bin3 family, which was first discovered in *Schizosaccharomyces pombe* [18].

Overall, it suggests that O-methylation of miRNAs constitutes a key modulator of ncRNA. The 2′-O-methylation methyltransferase BCDIN3D may be a potential target in the therapy of breast cancer. Numerous studies have highlighted the potential clinical importance of 2′-O-methylation by demonstrating that oral distribution of 2′-O-methylated miRNAs resulted in their absorption into intestinal cells and subsequent modulation of target mRNAs [19].

### 1.3. Environmental Epigenetic Influences

Epigenetic mechanisms have significant roles in changing gene activity states from one generation of cells to the next, which is a good reason why these mechanisms contribute to various diseases in humans. The environment and lifestyle influence epigenetic mechanisms such as histone modifications, DNA methylation, and miRNAs expression. Some of the epigenetic influences may positively affect the health of humans, for example, exercise, microbiome, and alternative medicine, or lead to changes in epigenetic regulatory features in the DNA methylation and miRNAs expression including environmental toxicants and drugs of abuse. Other factors may affect human health positively or negatively, depending on the nature of the impact, such as financial status, psychological state, diet, social interactions, seasonal changes, therapeutic drugs, and disease [20].

### 1.4. Epigenetic Modifications on Chromatin

In the mammalian genome, DNA methylation regularly happens from cytosine “5” to guanosines, and the 5′—C—phosphate—G—3′ (CpG) dinucleotide has been dynamically exhausted from the eukaryotic genome during evolution. Enhancing chromatin remodeling chaperon activity induces late replicating of the genome and thus plays a role in suppressing transcription of repeat regions, such as *Arthrobacter luteus* (Alu) restriction endonuclease sequences and transposons. Accordingly, the most widely investigated epigenetic mark involves the covalent transfer of a methyl group on the cytosine ring of DNA at position 5 (5mC) [21]. DNMT enzymes carry this label, normally in CpG sequences [22]. DNA methylation is among the primary epigenetic changes important to gene expression regulation in normal cell growth as well as cell differentiation [23]. Although 5-methyl cytosine is the best-known modification in DNA, other cytosine modifications have already been identified, such as 5-formylcytosine 5fC, 5-carboxylcytosine 5caC, and 5-hydroxymethycytosine 5hmC [24]. The function of these extraordinary markers is yet unknown. Nevertheless, 5hmC has a critical role in the central nervous system, where it is spread throughout the body, as well as in balancing pluripotency in undifferentiated cells [24]. 

Another major category of the epigenetics tag is the PTMs in histone proteins, which are responsible for packing long DNA molecules into nucleosomes [25]. Histone proteins are known as building blocks of chromatin, where 146 bp of DNA is wrapped around each octameric histone protein at about 600° [26]. There are five primary categories of histone proteins, (H2A, H2B, H3, H4, and H1). Histone fold domain proteins (HFDs) play critical roles in eukaryotic cells by heterodimerizing in specific pairs, H3 with H4 and H2A with H2B, which are linked by four-helix bundles, H3 with H3 and H4 with H2B. Thus, each octamer typically contains two H2A-H2B dimers and one H3-H4 tetramer, and H1 fills in as the linker histone protein between nucleosomes [27]. The HFDs with unstructured tails have different sites of PTMs [28]. V. G. Allfrey discovered acetylation modification of histone proteins in the 1960s, but we now know that a wide range of other histone modifications can occur. In 1997, the effect of some PTMs on chromatin structure was discovered using the crystal structure approach and a high-resolution X-ray technique of the nucleosome. The nucleosome was observed as a globular core with a flexible N-terminal outward in the obtained structure. The tails of histones are the most easily available targets for enzymatic activities downstream of various signaling pathways. In addition, they contribute to inter-nucleosomal interactions. Furthermore, this method influences the overall chromatin structure. Thus, it can be said that histone modifications affect transcription and many other DNA-mediated processes, such as recombination, repair, and replication [29]. H3 and H4 proteins have long tails that can be far from the nucleosome and can be covalently modified in some positions, while other histones can also be modified though to a lesser extent. The most known modifications are mono-, di-, and trimethylation, acetylation, and phosphorylation, although increasing modifications are also being recorded [30]. 

Traditional nomenclature shortens the histone protein, the changed residue domain, and the form of modification, so that acetylation in histone 3 lysine 27 is shown as “H3K27Ac”. The types of modifications that can occur in histone tails include acetylation, methylation, phosphorylation, ADP ribosylation, deimination, isomerization, ubiquitination, parylation, citrullination, and sumoylation [30]. These changes are revealed by a specific enzyme, such as HATs and HDACs for acetylation points, or HMTs and demethylases HDMs for methylation marks [30]. In comparison to histone modification, histone variants can have important transcriptional regulatory functions. Histone variations override canonical histones in order to change nucleosome architecture and, essentially, DNA availability [31]. An example of a histone variant is H2A.Z, which substitutes nucleosomal H2A to play multiple diverse regulatory functions in gene expression and evolution [32]. Eventually, this paper will include references to polycomb epigenetic repressors and bromodomain-containing proteins as well. Polycomb proteins can reshape chromatin and usually act as epigenetic gene silencers, whereas bromodomain proteins are amplifiers of the histone acetylation signal [33]. A person’s epigenetic profile could vary from one organ to another due to genetic differences and requirements. One of the best examples of methylation is the polymorphic CpG site [34]. If the substitution of a nucleotide breaks down the CpG of certain individuals, those individuals do not show methylation at the locus. Many different types of disease-associated polymorphisms have been discovered in CpGs, indicating that they contribute greatly to differences in human disease risk [35]. In this regard, human variations in epigenetic states can lead to disease, irrespective of whether they arise from genotype or environmental factors.

### 1.5. Application of Epigenetics in Personalized Medicine

Although each eukaryotic cell has a specific epigenetic pattern, individual people have an epigenetic profile in their cells that makes up their own epigenome. Epigenetic abnormalities are known to play a vital role in a couple of human disorders. The ultimate goal of custom-made epigenetics is to design diagnostic and treatment systems based on each person’s epigenetic profile. Individual epigenome variation is the foundation of personal epigenetics, which is epigenetics that is unique to an individual. DNA methylation, for instance, varies among people, as do several different ways of epigenetic histone modification [36,37]. Moreover, ncRNAs patterns are often variable from person to person [38]. These changes in epigenetic activity will inevitably become much more significant as the scope for precision epigenetics in medicine continues to expand. Nowadays, personalized epigenetics can act as a reference not only for the treatment of epigenetic disorders but also for many other areas of medicine. In addition, epigenetic risk factors, consisting of epigenomic signatures specific to each organism, have progressively been shown to give not just valuable data on the diagnosis of diseases and syndromes but also prognostic data on the conceivable non-occurrence or reappearance of life-threating diseases such as cancer [39]. To achieve the aim of precision medicine, the current task is to know how to acquire an accurate, useful biomarker for clinical routine and, for this reason, the new biomarker demands high precision and robustness [40]. It should be remembered that fewer than 1% of biomarkers collected in biomedical research are eventually placed into the clinical laboratory, with a still smaller proportion of epigenetic biomarkers [41].

The investigation of the genome and its related information can reveal insight into various issues related to an individual’s health and illness. Whole-genome DNA sequence details are now possible as a result of the culmination of the Human Genome Project (HGP). Special medications must be used by patients who do not adapt to conventional therapies as planned and for whom the rate of effective disease control is very low. Genomic or customized drugs are prescribed to patients after the processing of genomic information and related details, such as RNA levels, proteins, and different metabolites, which are important factors in medical decision-making for personalized medicines [42]. Genomic methods including the detection of DNA sequence variants, as well as transcriptomic, proteomic, and metabolomic methods are critical for efficient disease control and prediction [42]. These strategies are important instruments that make connections between epigenetics and personalized medication: human genome successions (genomics) contain 10–15 million Single Nucleotide Polymorphisms (SNPs) and Copy Number Variants (CNVs); gene expression patterns (transcriptomic) comprise around 25.000 gene transcripts; proteomes (proteomics) contain roughly 100,000 unique protein products [42]. SNPs play a significant role in regulating cell cycle regulation, DNA mismatch repair, metabolism, and immunity. SNPs are known as quite possibly the most regular kinds of hereditary variation in the human genome, and they have been identified as the strongest risk factors for various kinds of disease complexes such as cancer [43]. In Figure 2, three types of the main epigenetic marks: DNA methylation (caused by DNMT1-3), histone modification, and chromatin remodeler enzymes have been illustrated.

## 2. Epigenetic Abnormalities in Human Cancer and Brain Diseases

### 2.1. Abnormal Expression of DNA Methyltransferases (DNMTs) in Human Cancers

DNA methylation has significant roles in gene expression, chromatin stability, and genetic imprinting, and this process occurs via DNA methyltransferases (DNMTs), including DNMT1, DNMT3A, and DNMT3B. In addition to the previously mentioned enzymes required for mammalian DNA methylation, three new members of the DNMT family have been identified: DNMT3C, DNMT2, and DNMT3L. DNMT3C has a high level of similarity to DNMT3L and plays a role in the methylation of the young retrotransposons. DNMT2 has a role in the methylation of small transfer RNAs (tRNAs), which has not yet been detected in the catalytic activity of this enzyme, and DNMT3L, in the form of DNMT3L-DNMT3A heterotetramers, facilitates the methylation of cytosine residues [44,45]. Different genomic landscapes of human tumor tissues have been reported in recent years as a result of advances in next-generation sequencing technology, as have a number of defective and mutated genes linked to illnesses such as DNMT3A, TET2, and IDH1. These genes are directly or indirectly associated with DNA methylation and were identified as epigenetic processes. DNMT enzymes play critical roles in target gene transcription inhibition and catalysis, and chromosomal homeostasis and DNMT dysfunction are considered epigenetic disorders in diseases such as cancer [44]. The genetic structure of various DNMTs is shown in Figure 3. 

Based on previous studies, it seems that a part of the abnormality in DNA methylation pattern in human cancer is associated with mutations in DNMTs. Apparently, any changes in the expression of DNMTs have been influenced by human cancers. For instance, the rate of mRNA expression from DNMT1 is substantially higher in non-cancer liver tissue samples than in chronic hepatitis or cirrhosis samples [46]. The chance of overexpression of DNMT1 in Holocytochrome C Synthase (HCCS) is highly related to the weaker differentiation of the tumor and the presence of a vein tumor. Furthermore, overexpression of DNMT1 in tumors is indirectly related to recurrence-free and generally good endurance in HCC patients [47]. Adenocarcinoma cancer in the pancreas can be stimulated and spread again after the trauma of pancreatitis. Peripheral pancreatic ductal epithelium cells that have inflammatory or allergic properties can be in a precancerous state.

The effect of DNMT1 on protein expression has been studied since its function in peripheral pancreatic ductal epithelia has an acute inflammatory history. This can lead to precancerous tumors, pancreatic intraductal neoplasia (PanIN), extremely differentiated ductal adenocarcinoma, and potentially malignant ductal adenocarcinoma [48]. The high expression level of DNMT1 in ductal adenocarcinomas of the pancreas is highly related to the ability of invasion of cancer cells to the adjacent tissue, which almost occurred in an advanced stage with a lower patient survival rate [48]. The expression level of mRNA and protein from the DNMT1 gene is highly associated with weaker differentiation of the CpG Island Methylator Phenotype (CIMP) in stomach cancer. However, there was no correlation observed in DNMT3A, DNMT3B, or DNMT2 [49]. Infection with the Epstein–Barr virus in stomach cancers is greatly connected with the accumulation of DNA methylation on the C-type CpG islands. Besides, overexpression of the DNMT1 protein in infections with *Helicobacter pylori* is another causative factor that highly increases localized DNA hypermethylation [50].

Considering the mechanism that controls the level of expression of DNMTs [50], it has been proven that members of the miR-29 family, such as miR-29b, miR-29c, and miR-29a, are particularly regulated by DNMT3B and DNMT3A [51]. Increasing the expression of miR-29s in lung cell lines results in the correct DNA methylation profiles, as well as the expression of tumor methylation silencer genes and tumorigenicity in vitro and in vivo [51]. Since the decrease in expression of DNMT3B and DNMT3A is the result of a direct interaction of miR-29b with the 30UTR, miR-29b indirectly regulates DNMT1 expression by targeting the Sp1 gene, the DNMT1 gene trans-activator [52]. It has been shown that in human cancer cells, MiR-148 binds to DNMT3B and subsequently modifies DNMT3B splicing [53]. Moreover, miR-148 and miR-126 directly control DNMT1 expression and activity [54]. Decreasing miR-152 expression causes an undesirable DNA methylation pattern in HCC cells by altering DNMT1 levels. Hu-antigen R (HuR) proteins have recently been discovered to interact with target mRNAs and alter their expression levels by changing their balance. 30UTR DNMT3B is a HuR protein target in human colon cancer cells, caused by DNA hypermethylation of their target genes [55].

Conservation of the gene promoter DNA methylation correlates narrowly with DNMTs, which have key roles in human cancers. The mRNAs and proteins that encode DNMTs (DNMT1, DNMT3A, and DNMT3B) with their related catalytic activities are overexpressed in different cancer types, leading to aberrant behavior compared with normal cells. For example, important elevations of DNMT3B and DNMT1 expression have been associated with tumorigenesis, particularly in colon cancer (CC) [56]. 

Epigenetic changes via aberrant modifications of DNA and histones lead to modulation of the capacity of the genome to store and inherit genetic information, which significantly differs between normal and tumor cells. Abnormal DNA methylation and distinct histone modification patterns due to aberrant activity of epigenetic modifiers are very common in tumor cells and have effects on drug response and tumor growth. Cancer genes are classified as epigenetic modifiers, mediators, or modulators using epigenetic functional classification [56]. Epigenetic modifiers change the epigenome directly through DNA methylation, the modification of histones, or the alteration of the structure of chromatin via different genes (mutated or not) such as Smarca4, PBRM1, ARID1A, ARID2, ARID1B, DNMT3A, TET2, MLL1/2/3, NSD1/2, SETD2, EZH2, and BRD4. Furthermore, for epigenetic modulators and mediators, several genes have been identified (IDH1/2, KRAS, APC, TP53, STAT1/3, YAP1, CTCF, OCT4, NANOG, LIN28, SOX2, and KLF4) that modify DNA methylation or chromatin structure and play important roles in various types of cancer [57].

### 2.2. Relationship between DNA Methylation and Histone Modifications 

One of the pathways that determine the structure of chromatin and control gene expression levels is DNA methylation according to histone modifications [58]. Covalent histone modifications contribute to open promoters (H3K4 methylation and H3K27 acetylation), active enhancers activities (H3K4 methylation, H3K27 acetylation), the initiate of gene transcription (H3K36 methylation), and finally heterochromatin formation (H3K9 methylation, H3K27 methylation) [58]. MeCP2 and MBD2 as methyl-CpG-binding proteins bind to methylated CpG dinucleotides and transcriptional repression domains activate corepressor complex containing histone deacetylase (HDAC) enzyme. On the other hand, HMTs such as G9A and SUV39H1 are needed for DNMT enrollment. Transcription-suppressing chromatin shifts the tumor suppressor gene promoter to silence by DNA methylation and causes chromatin changes in healthy embryonic stem cells, e.g., the polycomb (PcG) binding domain. These genes seem to have an active bio-marker, e.g., H3K4 methylation in normal stem cells. Following stem cell differentiation, this bivalent gene function is converted into an active or repressive chromatin conformation structure [59].

In tumor formation studies, these modifications can give valuable information, bringing in abnormal DNA methylation identification. These PcG complexes are typically used to interact directly with DNMTs and may aid in the silencing of cancer-specific proteins such as enhancer of zeste homolog 2 (EZH2). PcG-binding proteins in the repressive polycomb complex, PRC2/3 catalyze trimethylation of H3K27, perhaps a key player in shutting down an upstream oncogene [60]. High EZH2 expression level is associated with tumor expansion and cancer invasion. Destruction of EZH2 in neoplastic cells can lead to the inhibition of tumor growth [60]. Chromobox protein homolog 7 (CBX7) is another PcG protein that has been shown to recognize repressive histone markers such as H3K9me3 and H3K27me3 [61]. Similar to the EZH2 protein, CBX7 can utilize DNA methylation machines for gene promoters and to improve quality silencing during tumor invasion. It has been a long time since scientists knew that specific diseases, such as cancer, comprise heterogeneous cell populations. A novel cancer stem cell hypothesis has shown that minuscule populations of reputed cancer stem cells, which are cells that stimulate malignancy or tumor-starting cells, can possibly cause a tumor. These cancer-initiating cells are often resistant to chemotherapy and radiation and are responsible for treatment failure. Additionally, they may be able to form metastatic foci in different organs or tissues. Although the existence of these populations, such as the cells in the cancer stem cell hypothesis, remains unclear, and the PcG complex identifies the same gene sets on cancer cells and embryonic stem cells, a more prominent effort should be focused on how epigenetic measures lead to the arrangement of cancer-causing cells [62].

### 2.3. Histone Modifiers in Cancer

Historical mutations in histone-modifying genes are extremely common in cancer, and some research and review articles concentrate on this subject [63]. In general, methylation of H3K4, H3K36, and H3K79 is associated with gene activation, while methylation of H3K9, H3K27, and H4K20 is associated with gene repression due to silent chromatin forms [64]. Since proteins associated with histone methylation (methyltransferases, demethylases, and methyl lysine-binding proteins) are deregulated in cancer, they have been studied as potential drug targets [65]. To demonstrate an illness mechanism, we briefly describe the H3K4 methylase and its role in cancer growth. The family of Mixed Linear Leukemia (MLL) diseases is caused by unregulated H3K4 histone methyltransferase genes and is often mutated in cancers. MLL disorder occurs more often in leukemia due to possible fusion gene formation, partial tandem replication, and duplication, and is thought to initiate tumorigenesis through capacity gain instead of loss of capacity. At times, the partner of combination epigenetic control genes, such as CREB-binding protein (CBP) and TET1 [66], is fused to the N-terminal of the MLL. Marker MLL genes are downregulated across MLL-associated leukemias, including HOX genes [67]. The HOX gene is an essential hematopoietic control system and a characteristic factor in cancer cells that plays a role in promoting regeneration. Different systems may likewise prompt MLL-mediated tumorigenesis. MLL protein, for example, can inhibit the transcription of Runt-related transcription factor (RUNX) and Core-binding factor subunit β (CBF), both of which are required biomarkers of hematopoietic differentiation [68]. Members of the MLL family are active in cancer research. Mutations causing more activation of MLL2 and MLL3 have been identified in non-Hodgkin’s lymphoma and medulloblastoma brain tumors [69]. In addition, MLL2 is involved with the most common mutated genes discovered by Lawrence et al. [70].

It has been proposed that control of H3K4 methylation is a fundamental mechanism for tumorigenesis. Surprisingly, mutations in H3K27me2/3 demethylase Lysine Demethylase 6A (KDM6A) are common in medulloblastoma [71]. Mutations in either the MLL2 or KDM6A genes can instigate a similar illness in Kabuki disorder [72]. Mutations in the (lysine-specific demethylase 1) LSD1 coding gene that acts as H3K4me1/2 demethylase are involved in various forms of cancer, as well as increased levels of the LSD1 protein that happen without genomic sequence changes [73]. Knockdown of LSD1 inhibited the spread of tumors, indicating LSD1 is a putative oncogene [74]. Although LSD1 is essential for cancer development [74]. LSD1 can also demethylate the P53 tumor suppressor [75]. The KDM5 family belongs to demethylase enzymes that remove H3K4me2/3 [76]. Nevertheless, it is not simply identified with mutations in the KDM5A or KDM5B genes, and overexpression of these genes in cancer-tumors can happen because of changes in the regulatory region or epigenetic patterns. KDM5A (also known as RBP2) is higher than normal levels in gastrointestinal [77] and lung cancer [78]. KDM5A particularly targets homeotic genes that control the differentiation process. Therefore, overexpression of KDM5A is assumed to reduce H3K4me2/3 level in the promoter genes involved in the genes responsible for differentiation and aging, as well as the inhibitory genes that cause cancer. KDM5A is also the binding partner of retinoblastoma (Rb) [79]. Rb can activate RBP2-inhibited genes, increasing or inducing differentiation. KDM5B (also called PLU-1) is raised in breast [80] and prostate cancers [81]. Cell growth is associated with KDM5B through its inhibitory activity on genes that regulate the cell cycle (growth or differentiation) [82]. Eventually, a mutation in KDM5C was confirmed in renal tumors [70].

### 2.4. Epigenetics and Brain Diseases

Specific changes in the chromatin environment could happen during old age and through age-related sicknesses such as neurodegeneration [83]. Notably, epigenetic variability between identical twins and monozygotic individuals increases over time [84]. The function of epigenetic interference in the development of age-related diseases including Alzheimer’s disease (AD) and Parkinson Disease (PD) is known to be associated with the observation of various phenomena in identical twins [85]. In addition, the neuroprotective effect of HDAC (reduction of oxidative stress in neurons) inhibitors in brain diseases suggests that epigenetic-caused diseases might contribute to enhancing the activity of the CBP/p300 signaling pathway [86]. Genetic variety, histone PTMs, DNA methylation, and genome arrangement have been connected to brain diseases in both animal models and patient tissues. It was reported that in the post-mortem brain tissues of identical twins, an abatement in the total amount of DNA methylation and hypermethylation in AD patients versus healthy controls [87]. Site-direct hypo- and hypermethylation on DNA of the AD patient’s brain were compared to controls in some researches [88]. Certain genes are subjected to alterations in chromatin modification that have functions in the pathogenesis of AD; for example, hypomethylation of genes engaged in the production of amyloid-peptide, inclusive of TMEM59 and PSEN1. The most recent study shows a broad loss of heterochromatin in transgenic animal models and humans affected by AD have significantly increased gene expression [89]. This phenomenon comes with increased oxidative stress and DNA damage, leading to the loss of heterochromatin and the death of neurons [89]. Synuclein α SNCA (-synuclein gene) reduced DNA methylation in irregular Parkinson’s disease patients versus controls in postmortem brain samples [90]. SNCA is a significant risk gene for PD that encodes α-synuclein and is a critical segment of Lewy’s body. A genome-wide assay of DNA methylation in blood and the brain identified hypo- and hypermethylated genes, such as those associated with Parkinson’s disease, with around 30% concordance between genes in different tissues [91]. In order to adapt to the change in methylation, the level of nuclear DNMT1 in PD postmortem brains has been reduced [92]. In rat neuronal cells, overexpression of α-synuclein shifts DNMT1 into the cytoplasm, proposing an elaborated system for diminished DNMT1 levels in PD [92]. Appealingly, the existence of a methyl donor group in DNA and histone methylation (surveyed as SAM/SAH in the blood) is related to PD symptoms. For example, improved psychological capacity was correlated with higher methylation content [93].

#### 2.4.1. Bipolar and Schizophrenia Disorders 

Information indicating the importance of regulation of DNA methyltransferase patterns in neuronal activity has been collected here from various studies. High concentrations of DNMT1 have been measured in γ-aminobutyric acid (GABAergic) neurons in the prefrontal cortex of patients with schizophrenia and/or bipolar disorders [94]. In these cell lines, the promoter regions of Glutamate decarboxylase (GAD) and RELN genes were hypermethylated and connected to low degrees of transcription in the two genes, suggesting that increased levels of DNMT1 were responsible for their downregulation. A decrease in GAD protein levels has been linked to a decrease in the inhibitory effect of GABA neurotransmitter and REELIN protein from cortical neurons, resulting in a decrease in dendritic spines and, as a result, deterioration of neuropil hypoplastic in pyramidal neurons. In fact, it has been proposed that reduced dendritic spine plasticity is responsible for the psychological impedance found in the patients with mania. Furthermore, it is conceivable that different genes associated with nerve cell activity may be altered due to elevated levels of DNMT1. Recently, both DNMT1 and DNMT3A in telencephalic GABAergic neurons have been shown to be dramatically increased in layers I and II of BA10 cortical neurons [95]. These findings could assist in clarifying the de-novo hyper-methylation pathways and the preservation of GAD and RELN methylation profiles in these neurons. Altogether, these discoveries show that extreme degrees of protection by methyltransferases may at least mostly clarify the morphology and capacity of abnormal neurons in schizophrenia patients.

#### 2.4.2. HSAN1 Disorder

Hereditary Sensory and Autonomic Neuropathy type 1 (HSAN1) disorder is a neurodevelopmental disorder recently proved to be engendered through mutations in the DNMT1 gene in humans [96]. HSAN1 is one of the human disorders in which genetic methylation defects are associated with neurodevelopmental disorders. Interestingly, HSAN1 is acquired through an autosomal dominant. Mutations in the DNMT1 region result in premature mutant proteins, diminished enzymatic activity, and a decreased heterochromatin binding rate during the G2 cell cycle, resulting in genome-wide hypo-methylation and local hyper-methylation. The well-known characteristic of autosomal dominant is especially interesting; wild-type proteins might be active in cells while not having the ability to perform methylation in the presence of mutant protein. For better understanding, this is the first report where a heterozygous DNMT1 mutation happens in a mouse or a human and the biological phenotype is changed. The molecular mechanism of disease in heterozygous HSAN1 includes haploid deficiency and dominant-negative effects. Since the HSAN1 mutation has taken place inside the DNMT1 domain, it could induce dimerization [96]. Thus, heterodimer formation between mutant DNMT1 proteins and wild-type readily contributes to the inadequate and/or inaccurate maintenance of methyltransferase function through dominant-negative mechanisms such as unsteadiness and disabled heterochromatin binding.

#### 2.4.3. Immunodeficiency, Centromeric Region Instability, and Facial Anomalies (ICF) Syndrome

DNMT3B dysfunctions are responsible for human Immunodeficiency-Centromeric instability and Facial defects syndrome (ICF), which affect multiple organs [97]. Mental retardation, chronic and persistent infections, facial dysmorphism, disorders in the skin and digestive organs, and chronic physical dysfunction with the loss of immunoglobulin A (IgA) are often the most common symptoms found in patients with ICF syndrome. Cytogenetic and molecular mechanisms of disease in such patients include chromosomal changes influencing the heterochromatic segments of chromosomes 1, 9, and 16 by translocation, chromatid chromosome fragmentation, somatic cooperation, and reciprocal translocation between homologous and non-homologous chromosomes. Heterochromatin at the pericentromeric region on certain chromosomes is often dislodged and merged to create multiradial structures in ICF syndrome. At the molecular level, the amount of methylation in patients with ICF disorder at the satellite DNA in the pericentromeric districts of chromosomes 1, 9, and 16, and in the long arm of the Y chromosome is less than that in normal people. The DNA sequence in the α satellite region has basically been unaltered. Studies utilizing cells from patients with ICF disorder have shown extensive hypomethylation, drawn-out replication time, nuclease hypersensitivity, and variable release of inert Y and X-chromosome genes. A broad protein expression pattern of lymphoblastic cell lines in three patients with ICF disorder and five normal controls showed critical contrasts in gene expression levels associated with signal transduction, mRNA transcription, immune response, growth, and neurogenesis. ICF cells comprise loss of methylation in the promoter area of many genes, such as LHX2, deficiency of histone H3K27 trimethylation (hindrance mark), acquisition of H3K9 acetylation, and H3K4 trimethylation (an activation mark). Many distress genes contain homeobox genes (such as the HOX gene) that are essential for the development of the immune system, brain, and cranial face. These genes showed a clear absence of binding affinity for the SUZ12 motif in the polycomb PRC2 inhibitory complex [97].

#### 2.4.4. Rett Syndrome

Regardless of whether DNA methylation is heritable or not, one of the significant functional consequences is the adverse effect on gene transcription. Methyl-CpG-binding domain (MBD) proteins in Rett disorder assume a significant function in this disease’s development. Rett syndrome is a regular neurodevelopmental disorder with an inherited deformity in epigenetic machinery. This syndrome is an X-linked neurological disorder based on a genetic defect in MeCP2, occurring in one girl out of 10,000 to 15,000 births. Girls who are affected by Rett disorder grow regularly until the age of somewhere between 6 and 18 months. The period of autism starts with an absence of psychological, motor, and social abilities. The diagnosis of emotional symptoms such as hand-to-hand clapping and hand wringing indicates a deliberate lack of motor function at this stage. Rett syndrome is a chronic disorder, and after a period where mental indications have all the earmarks of being steady, further weakening happens, prompting significant mental disability and motor impairment, including quakes, apraxia, and ataxia. A few boys have been recognized as having Rett’s condition, in which they usually grow up with a more serious disease than girls [98]. These results are consistent with a female X-linked mosaic of a mixture of normally functioning cells (X chromosome with mutant MeCp2 allele inactive) and defective cells (wild-type MeCp2 allele inactive). Each cell has MeCP2 protein activity. Boys are generally more influenced than girls, as they have a unique, imperfect Mecp2 allele. The sole Mecp2 mutation seems to have some impact on phenotypic expression; mutations in the amino-terminals of proteins are associated with the more elaborate clinical appearance related to mutations close to the carboxyl-terminus [99]. Nevertheless, a tendency for X-chromosome inactivation is more closely correlated with the severity of the disease. Certain individuals whose neurons exhibit X-chromosome inactivation with the mutated MeCp2 allele have a milder phenotype of Rett syndrome. Indeed, even some female carriers’ genes have been accounted for when an exceptionally mutilated profile of X-brokenness has brought about a high number of cells with the MeCp2 allele, an inactive mutant [100].

#### 2.4.5. ATR-X Syndrome

MeCP2 associates with a variety of other proteins [101] such as chromatin proteins, and defects in certain of the genes that encode these proteins are often linked with neurodevelopmental disorders. In molecular studies, MeCP2 is associated with α-thalassemia and the mental retardation condition. X-connected homolog protein (ATR-X) is a member of protein family of sucrose non-fermenting 2 (SNF2) chromatin remodeling factor. This protein utilizes energy from the hydrolysis of ATP to annihilate the stability of the nucleosome. ATR-X syndrome is a mutation of the ATRX gene under X-chromosome heredity. The syndrome has been restricted to men. In females, no clear physical symptoms have been found. Clinically, this syndrome is defined by α thalassemia, as well as a severe mental impediment caused by an extraordinary anatomical abnormality and varying degrees of genitourinary disorders. ATR-X has been linked to the regulation of histone H3 and DNA methylation, while mutations in the *ATRX* gene may lead to the downstream epigenetic and transcriptional effects. The most significant methylation changes in the 14 genomic loci provide a unique epigenetic signature for this syndrome. Phenotypic issues are not restricted to the nervous system since there are neurological anomalies, for example, craniofacial impairment, structural instability, and problems with the liver, kidneys, and intestines. Mild cerebral atrophy is detected, and incomplete or complete agenesis of the corpus callosum is recorded in two cases [102].

#### 2.4.6. Cornelia de Lange Syndrome

MeCP2 also associates with the underlying spaces of the cohesin-complex molecule. At that point, MeCP2-ATRX—cohesin involves regions of regulation in forebrain cells [103]. Given these molecular interactions, it is not astounding that genetic mutations in the genes that make cohesin regulatory proteins prompt neurodevelopmental irregularities. Cornelia de Lange Syndrome (CdLS) is a predominantly hereditary facial disease with particular characteristics such as gastroesophageal dysfunction, growth retardation, upper extremity malformations, and cognitive developmental abnormalities. Mental retardation in CdLS patients, while usually mild to extreme, shows a wide variety of variations. Most CdLS cases contain a single point mutation in the cohesin regulatory domain of the Nipped-B-like protein (NIPBL) or structural subunits structural maintenance of chromosomes protein 1A (SMC1A) and additionally SMC3 [104]. Mutations in NIPBL include the homolog of the yeast Scc2 protein in vertebrates, which is the regulatory domain of cohesion. These mutations mediate approximately 50 percent of the reported CdLS cases. The assessed results of NIPBL mutations are abbreviated or untranslated proteins, showing that NIPBL haploinsufficiency brings about CdLS phenotypes. Haploinsufficiency is a result of CdLS and is endorsed by a child with an enormous defect in the NIPBL area and outrageous CdLS signs [105].

#### 2.4.7. Rubinstein–Taybi Syndrome

Rubinstein–Taybi syndrome (RSTS) is an ultra-rare hereditary problem that can be distinguished by a number of phenotypic defects. In this disease, besides the nervous system, numerous organs are affected. The inherited pathway is autosomal dominant, and the main phenotype is characterized especially by behavioral delays. There are physical anomalies that were observed as well, including delays in development, which are frequently connected with unreasonable weight gain in later adolescence or pubescence, an unpredictable craniofacial appearance, and an increased danger of cancer growth. De novo mutations in CBP are associated with the syndrome in approximately 55% of cases. The genetic cause has not been detected in about 42% of patients. CBP belongs to HAT, which is strongly associated with the p300 protein family as well as is a transcriptional coactivator that has been shown to be associated with over 300 transcription factors and common transcriptional mechanisms [106]. CBP and p300 have been shown to control hematopoietic stem cell differentiation [107]. CBP deficiency occurs in intellectual disabilities related to mature disorders, according to studies on the pattern of inheritance of this disease in a subject’s mouse [108]. As far as its roles in neural growth are concerned, the lessening of CBP protein levels in the uterus prompts a reduction in the adequacy of neurogenesis and gliogenesis in cortical progenitor cells [109]. CBP has been proven to bind to neural promoters and induce histone acetylation and is, along these lines, known to be necessary for the expression of nerve-specific genes during nervous system differentiation. As a result, these findings are consistent with previous claims that neurological conditions are “less severe outcomes” of slightly abnormal epigenetic machinery [109].

#### 2.4.8. Coffin–Lowry Syndrome

Coffin–Lowry syndrome (CLS) is one of the rare neurodevelopmental disorders that affects chromatin remodeling and correction of chromatin design defects. This is an X-linked condition observed in a variety of significant structural disorders in male patients, including neurodevelopmental disorders. At the molecular level, Coffin–Lowry syndrome is caused by loss-of-function mutations in RSK2 (also known as RPS6KA3), which encodes a serine/threonine protein kinase in humans. Coffin–Lowry is an epigenetic disease because RSK2 normally influences chromatin structure through two different mechanisms: direct phosphorylation of histones and by interacting with CBP, a histone acetyltransferase. Phenotypic manifestations in male patients include significant delays in physical growth and psychomotor development, whole-body hypotension, and skeletal deformity and inability [110]. 

#### 2.4.9. Kabuki Syndrome

Kabuki disease is characterized by mental disorders, short stature, specificity in facial expressions, developmental delay, and genitourinary defects. Recent studies have identified genetic mutations in the autosomal MLL2 gene in almost half of the patients with Kabuki syndrome [111]. Either of these cases was caused by prevalent de novo MLL2 mutations. Mutations have been identified in the gene that include splice site mutations, minor deletions or insertions causing frameshift mutations, and nonsense mutations. MLL2 encodes a large 5262 residue protein that is part of the SET protein family, Trithorax, in the Drosophila homolog of MLL. The SET domain of MLL2 provides the heavy activity of histone 3 lysine 4 methyltransferase and is essential for the epigenetic regulation of chromatin-active states. Many of the MLL2 variants that have been found in individuals with Kabuki syndrome are expected to break the polypeptide chain before the translation of the SET domain. The disease is more likely to arise due to haploid rather than functional gain because some pathogenic bias variants are found in the MLL2 region encoding the C terminal domains. Therefore, this Kabuki syndrome in living people with MLL2 mutations is expected to manifest only as a partial loss of function [111].

## 3. Epigenetic Therapy 

Epigenetic modification is involved in the development of a number of diseases, and the knowledge needed to modify these systems can be used to treat many diseases [112]. Epigenetic drugs are generally chemical compounds that correct PTMs of histones and DNA [113]. In general, epigenetic drugs are classified into five groups: DNA methyltransferase inhibitors DNMTi, histone methyltransferase inhibitors HMTis, histone demethylase inhibitors HDMis, histone acetyltransferase inhibitors HATi, HDACi, and miRNAs. Among the five categories, DNMTi and HDACi are broad re-programmers that lead to general alterations in the epigenome. Other inhibitors are used for specific genetic changes in epigenetic pathways, such as the EZH2 inhibitor (Target Therapies) [114]. 

A. DNMTi binds to the DNA methyltransferase enzyme by covalent bonding and prevents it from binding to DNA by occupying the active site of the enzyme. They irreversibly inhibit DNA methylation [115]. When different lines of cancer cells are exposed to certain doses of these inhibitors, they could inhibit cell cycle progression as well as tumor spread [116]. In addition, DNMTi could reactivate tumor suppressor genes that had been silenced by abnormal methylation [117]. By using DNMTi, dormant antigens such as Cancer Testicular Antigen (CTA) can be increased in malignant cells and subsequently activate anti-tumor immunity [118]. CTAs are displayed in early embryonic cells but are usually silenced in adult somatic cells. Mutations in DNMTs, especially the enzyme DNMT3B, can cause excessive methylation, and therefore the use of DNMTis can induce the immune effects mentioned above [119]. According to Figure 4, examples of important DNMTis are given here: 

A-1. 5-azacytidine (Vidaza); the drug is approved by the FDA for the treatment of MyeloDySplastic (MDS) [120]. The functional mechanism of this drug is that by binding to the enzyme DNMTs, it inhibits DNA methylation. At low doses, this drug reduced DNA methylation in cell culture and resulted in the formation of cardiac muscle cells from mouse embryonic cells; this indicates that this drug is a simple cytostatic drug and in low doses, in addition to being non-toxic, induces phenotypic changes [121,122]. 

A-2. 5-aza-2′-deoxycytidine (Decitabine); which is functionally similar to 5-azacitidine but chemically has one less oxygen in its structure than 5-azacitidine. Decitabine binds to intracellular deoxyguanosine via phosphodiester bonding. This drug is a stronger methylation inhibitor than 5-azacitidine [123]. Decitabine has shown strong anti-proliferative effects in the face of ovarian malignant cell lines [124]. Immunologically, Decitabine induces natural killer cells (NK cells) proliferation and can also differentiate naive T cells into effector T cells (CD$8^{+}$ T-cells), which also secrete cytokines such as interferon-γ (IFN-γ) and tumor necrosis factor-α (TNFα), eventually causing cancer cell death [125]. Decitabine is a nucleoside analog that exhibits anticancer activity when it is incorporated into DNA and forms an irreversible covalent complex with DNA (cytosine-5-)-methyltransferase 1 (DNMT1), which leads to the degradation of the enzyme and consequently, the hypomethylation of aberrantly hypermethylated promoters [126]. 

A-3. Zebularin; has been shown to have effective anti-proliferative activity on ovarian malignant cell lines [127]. It has a similar mechanism of action to previous drugs, but in its chemical structure compared to 5-azacitidine, the amine group has been removed, and this deletion of the amine group has made it resistant to intracellular deaminases. 

A-4. Guadecitabine (SGI-110); It is a dinucleotide consisting of guanine and decitabine. Due to its structure, this drug has minimal toxicity and a longer half-life due to its resistance to intracellular deaminases. Experiments have shown that the drug can act as a prodrug and trip to the target tissue, which is the malignant tissue of the bladder, where it is degraded and activated by phosphodiesterase, and then to the promoter of the p16 gene, which is a suppressor gene. The target has been inactivated due to improper methylation, affecting, and reactivating it. It has also been shown to be effective in treating acute myeloid leukemia (AML), MDS, and liver cancer [128,129]. SGI-110 has recently been used to restore the sensitivity of Irinotecan in cases with colorectal carcinoma [130]. SGI-110 is much more potent in increasing dormant antigens such as CTA than 5-azacitidine [131]. Another inhibitor from this family is SGI-1027 can act in micromolar doses as a potent proliferation inhibitor in breast cancer, prostate cancer, histiocytic lymphoma, and Burkett lymphoma. This drug can also activate MLH1 and P16 promoter regions in colon cancer by inhibiting DNMT [132].

A-5. A new finding in DNMTi is the production of oral Decitabine (ASTX727), which combines Decitabine with the cedazuridine (cytidine deaminase inhibitor or E7727). This strategy has improved bioavailability [133]. Experiments have shown that several drugs that have been used in the past for non-epigenetic purposes, such as the antiarrhythmic drug procaine amide, epigallocatechin gallate, and the antihypertensive drug hydrolase, have been found to have inhibitory DNA methyltransferase properties and can inhibit DNA methylation [134,135]. Procainamide and procaine can modify CpG regions of DNA and ultimately block DNMT activity. It is also reported that flavonoids and EGCG can inhibit DNMT1 enzyme activity to restore RXRα expression in human colon cancer cells [136]; although they are less effective than drugs like Decitabine. The list of these drugs is collected in Table 1. 

B. The second class of epigenetic drugs is HMTis.

Many HMTis are in the investigation phase and have not yet been approved by the FDA. Among HMTs, EZH2 and Disruptor of telomeric silencing 1-like (DOT1L) are good targets for epigenetic therapy. 

B-1. EPZ004777 is one of the first competitive inhibitors to alter DOT1L expression. Pinometostat (EPZ5676), the second generation of these drugs that selectively inhibit H3K79 methylation, has better pharmacokinetic properties than the previous drug and is being studied and tested for the treatment of diseases such as MLL and AML leukemia [138,139]. Treatment with pinometostat by inhibiting H3K79 methylation causes the expression of MLL-fusion genes in patients with AML [140].

B-2. EPZ015938 is a selective inhibitor of PRMT5. This inhibitor is now undergoing clinical investigation for patients with solid tumors and non-Hodgkin’s lymphoma [141].

B-3. GSK-126 was the first EZH2 enzyme inhibitor to be discovered. This drug is being used in clinical trials to treat lymphoid malignancies and myeloma [142]. 

B-4. Another inhibitor of EZH2 is 3-Deazaneplanocin A (DZNep), which alters methionine metabolism; it degrades EZH2 and inhibits H3K27 methylation, thereby inducing apoptosis in MCF7 breast cancer cells and HCT116 colorectal cancer cells [143].

B-5. Yuan et al. showed that BRD4770, which is a G9a inhibitor, together with gossypol causes the death of cancer cells in pancreatic cancer [144].

B-6. UNC 1999, a SAM-competitive dual inhibitor of EZH1/2, inhibited MLL-rearranged acute leukemia cell proliferation [145].

B-7. BIX-01294 and UNC0638, the first selective G9a inhibitor, and its progressive alternative, are potential prospects as antitumor agents [146].

B-8. Tazemetostat (EPZ-6438), which is an oral EZH2 inhibitor that can be used to treat B-cell lymphoma. Tazemetostat was approved by the FDA as the newest epigenetic drug in 2020 for the treatment of advanced epithelioid sarcoma [147].

C. The third group of epigenetic drugs are HDMis, which are similar to histone methyltransferase inhibitors, are still in the research or study phases and have not yet been approved by the FDA. Two important families in this group are: Lysine-specific demethylase 1 (LSD1/KDM1) and Jumonji of HDM, which is an important therapeutic target in this category. Small molecules that inhibit Lysine demethylase1 (LSD1) or Lysine demethylase1 (KDM1), such as ORY-1001 and Tranylcypromine, can increase H3K4 methylation and thereby inhibit tumor suppressor genes; these drugs can be effective in treating blood malignancies and other types of cancer. Isocitrate dehydrogenase 1 (IDH1) and isocitrate dehydrogenase 2 (IDH2) inhibitors can have broad-spectrum effects on Jumonji class demethylases; examples of this category include AGI-5198 and AGI-6780 [148]. Enasidenib and ivosidenib have been approved as IDH inhibitors for regressed or refractory AML [149].

D. Another class of epigenetic drugs are HATis. There are several small molecules in nature that have been shown to inhibit histone acetylation; these compounds include anacardic acid, garcinol, and curcumin [150]. 

D-1. Although garcinol and anacardic acid have similar targets and are used to treat cancer, the cellular permeability of garcinol is much higher than that of anacardic acid. When malignant ovarian cell lines were treated with garcinol and anacardic acid, garcinol caused apoptosis, whereas anacardic acid only sensitized these cells to ionizing state [112]. 

D-2. The next substance is curcumin, which is the main substance in turmeric and responsible for its yellow color. This substance inhibits CREBBP and EP300, which is a HAT and can inhibit the acetylation of P53 in vivo [151]. Curcumin can also exert its anti-cancer properties by regulating cyclin D, CASP8 (caspase-8), and inhibiting NF-κB (nuclear factor κB) secretion. It can be used to treat multiple myeloma, breast cancer, and prostate cancer [152,153].

D-3. BET proteins have been shown to be lysine-acetylated histone-binding proteins (as epigenetic readers) involved in the translation elongation phase. It may be involved in the process of cell cancer; therefore, inhibitors of this family, such as JQ1 (BRD4) inhibitors, can be a good treatment option for some cancers [154,155].

D-5. Another example of BRD inhibitors is Apabetalone (RVX208), which treats several cardiovascular diseases [156].

E. Other classes of drugs are histone deacetylase inhibitors, which are major constituents in suppressing abnormal genes in cancers [115]. As is common, HDACi can repress the cell cycle in the G1 or G2-M phase, which can lead to unusual differentiation as well as the induction of apoptosis. These inhibitors can also sensitize cancer cell lines to chemotherapy drugs by affecting angiogenesis and tumor metastasis [157]. A class of HDAC inhibitors may include natural compounds such as short-chain fatty acids and small cyclic tetrapeptides such as HDACis phenylbutyrate, sodium butyrate, and valproic acid. Moreover, newer and more selective classes of HDACis may be made of compounds such as hydroxamic acid (such as vorinostat, Panobinostat, Belinostat, Pracinostat, Dacinostat, and Trichostatin A), benzamides (such as Entinostat, Mocetinostat, and Rocilinostat), and bicyclic depsipeptides (such as Romidepsin) [112,158]. The main mechanisms of action of HDACi include cell cycle arrest by factors such as p53 and p21CIP/WAF1 [159]. Moreover, HDACis inhibits metastasis by decreasing the expression of genes involved in angiogenesis, migration, epithelial-to-mesenchymal transition, and cell survival, while increasing the expression of genes involved in apoptosis [160]. Below are some of the important drugs in this category.

E-1. Suberoylanilide hydroxamic acid (SAHA; vorinostat), which is an oral inhibitor of histone deacetylase. This drug has been approved by the FDA for the treatment of cutaneous T-cell lymphoma (CTCL) and is being investigated in some separate clinical trials for its effects on other cancers [115]. The target of this drug is HDACs (class I, II, and IV). It is known that this drug, in combination with cisplatin and paclitaxel, two common drugs in chemotherapy, can improve the immune response and increase survival in patients with Non-Small Cell Lung Cancer [115,161]. 

E-2. Belinostat (PXD101), which inhibits histone deacetylation through its sulfonamide-hydroxide structure, is approved by the FDA for treating cutaneous T-cell lymphoma (CTCL) and peripheral T-cell lymphoma (PTCL) [162]. It is a pan-HDAC inhibitor, meaning it inhibits all classes of HDACs [158]. 

E-3. Panobinostat (LBH589) is an oral HDAC inhibitor approved by the FDA to treat multiple myeloma (MM). It is currently the most potent histone deacetylase inhibitor on the market [163]. It also targets class I, II, and IV HDACs [163]. This HDAC inhibitor, in addition to inhibiting TNBC cell metastasis, can sensitize TNBC cells to treatment with PARP inhibitors and cisplatin [164]. All three drugs (Vorinostat, Belinostat, and Panobinostat) contain a hydroxamic acid part that can bind to the zinc atom, thus inactivating HDACs [165].

E-4. Romidepsin (Depsipeptide) is a cyclic tetrapeptide that reduces thiol release by forming a disulfide bond, thereby inhibiting histone deacetylation. It is FDA approved for the treatment of peripheral T-cell lymphoma (PTCL) and cutaneous T-cell lymphoma (CTCL) [166,167]. The targets of this drug are HDAC1 and HDAC2, which means that it is more specific than the previous drug [158]. Romidepsin is disulfide-bonded, which is reduced by glutathione to release a zinc-bound thiol in cells. Then, this thiol interacts with zinc ions in the active site of class I and II HDAC enzymes, and as a result, its enzyme activity is inhibited [165].

E-5. Valproic acid, which is a short-chain fatty acid, has been used as an antiepileptic drug from the past to the present. In addition to its anticonvulsant and anti-migraine properties, it can also act as an HDAC inhibitor; in other words, the targets of this drug can be GABA and HDAC1. The drug inhibits tumor growth and metastasis in the malignant ovarian cell line SKOV-3 by inhibiting HDAC1 [158,168].

E-6. Another HDAC inhibitor called MPT0E028 inhibits growth in B-cell lymphoma by inhibiting HDAC, Akt phos, P53, Myc, and STAT [169]. The results of a series of studies have shown that the combined use of HDAC and DNMT1 inhibitors has a profound effect on the sensitization of chemotherapy-resistant breast cancer cells that can be used as anti-cancer therapies [170]. HDAC inhibitors such as resminostat, pracinostat, givinostat, abexinostat, entinostat, and quisinostat are going through clinical trials [165].

F. miRNAs have been linked to many diseases, including cancer. One series of miRNAs acts as a tumor suppressor and another as an oncogene [171]. Changes in the expression of miRNAs can disrupt the regulation of the cell cycle and lead to changes in the cell, including adhesion, growth, invasion, and escape of the immune system, all of which can be involved in the process of a cell tumor [172]. Short and lncRNA molecules are disrupted in the tissues of patients suffering from brain diseases, so they are dramatically proposed as disease biomarkers as well as possible targets for therapeutic interventions [173]. In 2011, Yuan and colleagues found that HDACs suppress several miRNAs, such as miRNA-200a. Suppression of miRNA-200a in hepatocellular carcinoma plays an important role in metastasis [174]. miRNA-200a, as a tumor-suppressing miRNAs, can be reactivated by using HDACis [175,176]. 

Recent studies have also shown that HDACis increase cell death in chronic lymphocytic leukemia (CLL) by increasing the expression of some miRNAs, including miRNA-15 and miRNA-16 [177]. Mocetinostat (MGCD0103), an HDACi, leads to an increase in anti-tumor activity against stem cells in the early stages of prostate cancer by increasing miRNA-31 expression [178]. The combination of DNMTi and HDACi, in addition to regulating homeostasis in cancer cells, can increase the level of gene induction and expression of miRNAs and lead to independent activation of endogenous long tandem repeats (LTRs) [179,180]. Studies use both direct and indirect strategies for using miRNAs in the treatment of cancer: The direct strategy is divided into two categories; (A) Use of miRNAs antagonists to inhibit oncogenic miRNAs. (B) The use of tumor-suppressing miRNAs in the form of artificial miRNAs or as products of transfection of miRNAs-encoding genes by vectors such as adenoma virus-based vectors and plasmid vectors based on miRNAs expression [181]. The indirect strategy involves the use of drugs to modulate the expression of miRNAs by targeting transcription and regulating gene expression [181]. This category can include miRNA-nanoparticle compounds, miRNA sponges, extracellular vesicles (including two groups of microvesicles and exosomes), and miRNAs-Mask [181]. SMIRs are synthetic organic small molecules that can irreversibly bind to miRNAs. Mechanistically, they bind to grooves and pits on the surface of miRNAs and interfere with the biological function of target miRNAs [182]. Several studies have shown that epi-drugs can either increase or decrease miRNA expression. For example, garcinol, a HATi, inhibits the overexpression of miRNA-224 [183,184]. It is found that miRNA-34a acts as a tumor suppressor miRNA through Neurogenic Locus Notch Homolog Protein 1 (NOTCH1) and CD44 signaling to induce apoptosis and suppress tumor proliferation, migration, and growth in Triple-Negative Breast Cancer (TNBC) cells. The use of doxorubicin in combination with miRNA-34a may be effective in treating TNBC [185]. Today, most scientists focus on combining epigenetic and immunotherapy drugs; whether for diseases that have a specific immunotherapy drug such as multiple myeloma (MM) or diseases that do not have a specific drug, such as some cancers and many blood malignancies such as Acute Myeloid Leukemia (AML) and Myelo Dysplastic Syndromes (MDS) [186]. An HDACi called Trichostatin A reduces Bcl2 by increasing the expression of miRNA-15 and miRNA-16, thereby increasing survival in patients with lung cancer [187]. Complete inhibition of HDAC2 can increase miR-183 transcriptional activity by increasing histone H4 pan-acetylation in the miR-183 promoter region, leading to miR-183-mediated tumor suppression in neuroblastoma [188]. The list of these four groups of epi-drugs is shown in Table 2.

### 3.1. Combination Effects of Using DNA Methylation and Histone Modification Inhibitors for Epigenetic Therapies 

The investigation of the correlation between histone modification changes and DNA methylation aberrations (see Figure 5) has inspired researchers to use both therapeutic agents, combining DNA methylation inhibitors and HDACi. Jahangeer et al. have shown that using 5-aza-CR and butyrate (HDACi) has a cumulative effect on decreasing adrenergic receptor expression in HeLa cells [216]. Afterward, Ginder et al. reported that the two medications act additively in anemic chickens to elevate the measure of embryonic p-type globin messenger RNA in hematopoietic cells [217]. Cameron et al. studied the synergism of both methylation and HDACis. Yamashita and Suzuki et al. have utilized a way of treating cultured cancer cells with 5-aza-CdR and trichostatin A simultaneously to effectively separate new tumor-silencer genes [218,219]. Furthermore, a novel study showed a decent synergistic effect between 5-aza-CdR and phenylbutyrate an HDACi for the hindrance of murine cancer in the lungs [220]. This is a surprising time when clinical preliminaries keep having the option to screen for these two epigenetic transformations in patients [221]. Studies have shown that adjuvant 5-azacytidine and low-dose entinostat disrupt the pre-metastatic microenvironment and prevent the formation and growth of lung metastases. Thus, a combination of low-dose DNMTi and HDACi may be effective in cancer treatment by inhibiting the metastasis of solid tumors. It was also found that the combination of DNMTi (azacitidine) and HDAC6i (NextA) resulted in a type I interferon response and increased expression of cytokines and chemokines in human and mouse ovarian cancer cell lines [222]. Moreover, these techniques could be useful to treat the cells through a genetic treatment involving chemotherapy [223], interferon [224], or immunotherapy [225], among others. This process is shown in Figure 5. 

### 3.2. Potential Side Effects of Epigenetic Therapy

In addition to the benefits of epigenetic therapy, there are many questions about the therapeutic use of these agents. These are primarily concerned with broad-spectrum switching on of genes and transposition factors in normal cells, as well as their ability to induce mutations and carcinogenesis. Unfortunately, several experiments have looked at the effects of azanucleosides on perfectly normal cells instead of cancer cell lines. The drug had a dramatic effect on the immortal lines, but of the 6600 genes studied (compared to 1% of tumor cell lines), only 0.4% increased more than 4-fold in normal human fibroblasts exposed to 5-aza-CdR [226]. However, early experiments showed that the 5-AZA-CR can activate human chromosome X in the hybrid of somatic cells in rodents but not in normal human cells [227]. These results suggest that DNA methylation is the only pathway that induces silence in normal cells. Furthermore, they are less amenable to drug-prompted gene activation. Imprinted genes can be activated with 5-aza-CdR, but we need to take caution when implying it [228]. Azanucleosides have been shown in mice to be mutagenic and potentially cancer-causing agents, as well as to activate silencing in cancer cells [229]. However, it can act as a cancer preventive agent [220]. In clinical trials, it was discovered that azanucleosides have some specific benefits with no evidence of negative effects. For instance, treatment of leukemia disease in patients taking 5-aza-CdR shows some effects on overall genomic demethylation, as evaluated by shifts in Alu methylation [230]. Early methylation levels were reestablished within about fourteen days of treatment, and no growth of secondary cancer was found in follow-up examinations. Furthermore, a low dose of 5-aza-CdR-caused cell reduction when used in a variety of patients with myelodysplastic disorders who have preexisting chromosomal deformities (sex, age, etc.). No increment in chromosomal instability was observed during treatment, which goes against the strong effect of 5-aza-CdR on the patient’s chromosomal integrity. DNA methylation and the treatment mechanisms for HDACis are never straightforward. Apparently, all types of drugs can activate genes, so this is how they work with HDACis in patients and DNA methylation inhibitors, which are cytotoxic drugs and upregulate p21 and/or p53, which lead to cell cycle stop and cell death [231]. Deactivation of genomic methylation induces P53-dependent apoptosis, and P53 inhibits DNMT [232]. It can be proposed that there is an interaction between the two proteins. Specifically, cytotoxic effects can occur when proteins containing DNA methyltransferase bind to the DNA of azanucleoside-treated cells [233]. These factors make it important to test the patient’s alternative endpoints to gain a deeper understanding of how they work.

## 4. Discussion

The Human Genome Project (HGP) has developed thousands of new cancer therapeutic targets [234]. The HGP, on the other hand, did not explain the variations in gene expression that occurred throughout the course of cancer’s onset. The interplay between tumor suppressor genes and oncogenes is regulated by genetic and epigenetic alterations that contribute to carcinogenesis and metastasis. Unlike genetic mutations, which alter the genome’s sequence, epigenetic mutations influence gene expression [235].

Dysregulation of the epigenetically marked state, as well as disorders in epigenetic markers or enzymes related to these processes, can play important roles in disease development, and modulating such systems could be a therapeutic strategy in drug discovery. Epigenetic therapy is known as a significant treatment method in disease therapy, but it is still in its early stages. In recent decades, increasing knowledge about cancer biology and brain diseases has shown the importance of genetic aberrations and their roles in cancer cells. The study of the cancer epigenome leads to the detection of the biological pathways in cancer, and as a result, the development of novel epigenetic therapies and the use of epigenetic drugs to inhibit various enzymes by adaptor proteins involved in epigenetic diseases. Based on this study, we can say that aberrant DNA hypomethylation leads to more diseases and affects oncogene activation and silencing in many tumor suppressor genes. Thus, intensive sampling of methylated genes in cancer disease for the detection of tumor-related genes can be a significant therapeutic target. Furthermore, modifications in the expression of DNMTs play a key role in human cancers.

Mammal-specific short single-stranded noncoding RNA molecules called miRNAs are involved in posttranscriptional gene control and gene silencing. They have a 22-nucleotide length and are found in mammals [236]. Mechanically, miRNAs negatively regulate the gene expression of target mRNAs via the sequence-specific base-pairing of miRNAs with 3′ untranslated regions of target messenger RNAs, followed by the cleavage of the mRNA strand [236]. Because miRNAs are expressed in a cell-specific manner and play a role in biological processes, such as cell proliferation, differentiation, and apoptosis, aberrant miRNA expression plays a role in cancers of various origins, including the breasts, colon, stomach, lungs, prostate, and thyroid [237].

Specific types of miRNAs play tumor suppressor or oncogene roles, as do different types of their direct-control DNMTs. Nonetheless, down-regulation of miRNAs may cause abnormal expression of DNMTs in various cells and lead to different cancers. On the other hand, HDACis increase cell death by increasing the expression of some miRNAs. Several mechanisms, including chromatin, modifications, drug efflux, upregulation of oxidative stress response mechanisms, defects, or upregulation in apoptotic pathways, contribute to HDACi resistance. These hindrances can be overcome, at least in part, by combining HDACi with other anticancer drugs [238]. To sensitize cancer cells to chemotherapy drugs, epigenetic drugs are prescribed before or simultaneously with chemotherapy. The synergistic effects of epigenetic therapy maximize the effectiveness of chemotherapy drugs. It has also been shown that the combination of epigenetic therapy and immunotherapy can increase anti-tumor immune responses [239]. The combination of DNMTi and HDACi leads to the regulation of homeostasis in cancer cells, increasing the level of gene induction and expression of miRNAs. Synergistic effects were observed when combining HDACi with, for example, PARP inhibitors, topoisomerase inhibitors [240], proteasome inhibitors, antimetabolites, radiotherapy, mammalian target of rapamycin (mTOR) inhibitors, or monoclonal antibodies [238]. Consequently, epigenetic drugs as chemical compounds can play an effective role in the deletion of post-translational disorders of histones and DNA. On the whole, among the epigenetic drugs, DNMTi and HDACi are broad re-programmers that lead to general alterations in the epigenome and in epigenetic pathways. Numerous epigenetic therapies have reached clinical trials and only a few have been approved for patient use by the FDA.

Recently, HDACi and immune checkpoint inhibitors have been used in combination to treat various cancer types, and the results are encouraging. In this line, Knox et al. demonstrated that the combination of anti-PD-1 and ultra-selective HDAC6i Nexturastat A significantly improved antitumor immune responses [241]. According to this study, this combination therapy altered tumor development, along with tumor-infiltrated cells, and the cytokine microenvironment, making it more responsive to immunotherapy. In syngeneic melanoma tumor models, this treatment approach ultimately greatly reduced tumor growth. In a different melanoma model, HDAC inhibition was also demonstrated to enhance immunotherapy in triple-negative breast cancer [242], multiple myeloma [243], and B-cell lymphomas.

Furthermore, many complex multifactorial diseases of humans with novel drugs have not yet been effectively treated, and they are difficult to treat against many epigenetic regulatory mechanisms such as histone methylation/demethylation or acetylation/deacetylation. Therefore, more research is needed in this field in the future.

## 5. Conclusions

In cancer, alterations in the genome and epigenome are obvious and lead to mechanisms by which the growth of tumor cells is out of control, and surveillance and becomes increasingly independent of the host. Unlike genetic mutations, epigenetic changes exhibit a much greater degree of flexibility. Thus, epigenetic alterations have significant roles in immune surveillance and developing drug resistance. Epigenetic processes, particularly DNA methylation, are associated with gene mutations and contribute to stability in gene expression, which is somatically inherited. Therefore, tumors have the advantage of undergoing much faster evolution through genetic alterations alone.

Epigenetic drugs may prove useful in treating many diseases by inhibiting DNA methylation or aberrant histone acetylation or methylation. Multiple cellular signaling pathways are affected by epigenetic drugs, such as immune response and evasion, apoptosis, cell survival, and DNA damage repair. Epigenetic drugs are used for the inhibition of various enzymes such as DNMTs, HMTs, HDMs, HATs, and HDACs in epigenetic diseases and can effectively act in combination with other therapies such as standard chemotherapy or immunotherapy. In the future, epigenetic therapy may be developed as the most effective method for treating cancer and brain disease.

## Figures and Tables

**Figure 1 genes-14-00873-f001:**
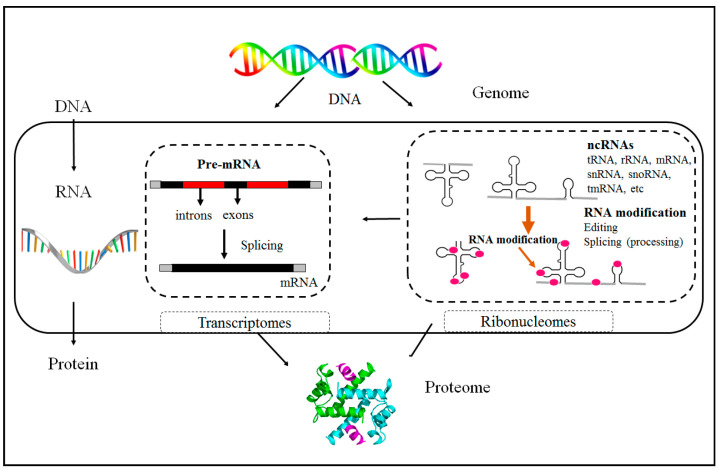
The relationship between epigenetics and genetics: As genetic information is transferred from DNA to RNA and then to protein, epigenetic information is applied as changes to DNA, RNA, and protein and can affect the function of the proteome.

**Figure 2 genes-14-00873-f002:**
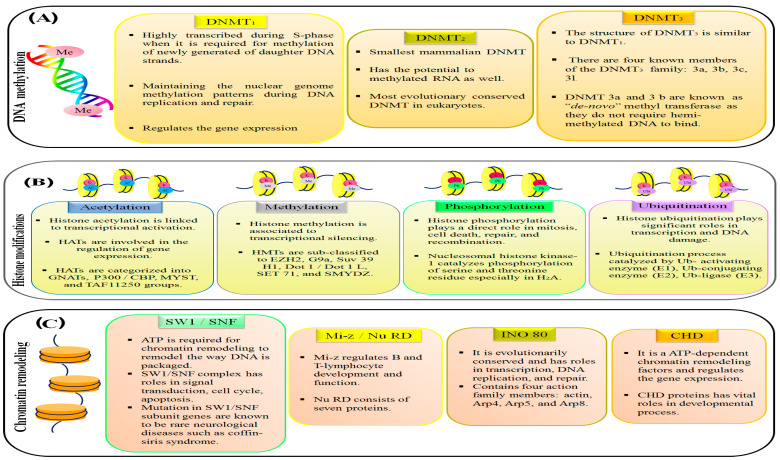
Selected examples of the possible interactions between the various epigenetic factors, such as DNA methylation (**A**), histone marks, (**B**), and nucleosome positioning (**C**).

**Figure 3 genes-14-00873-f003:**
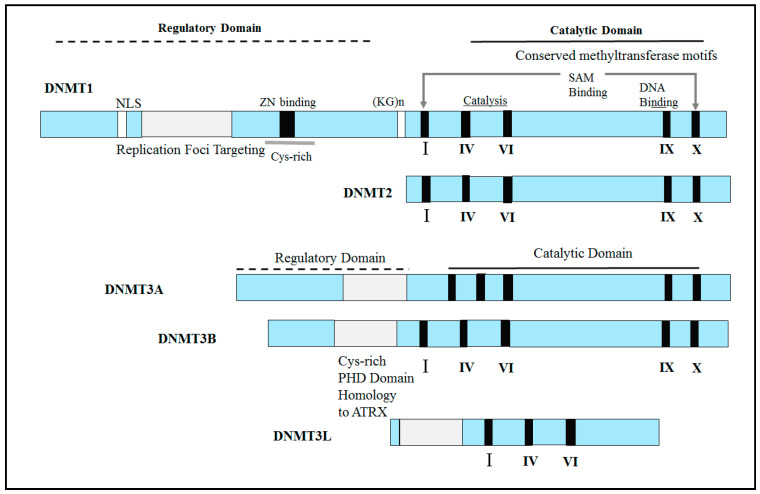
The structures of the known DNMTs contain two domains, regulatory and catalytic. Roman numerals refer to conserved motifs of DNA methyl transferase [45].

**Figure 4 genes-14-00873-f004:**
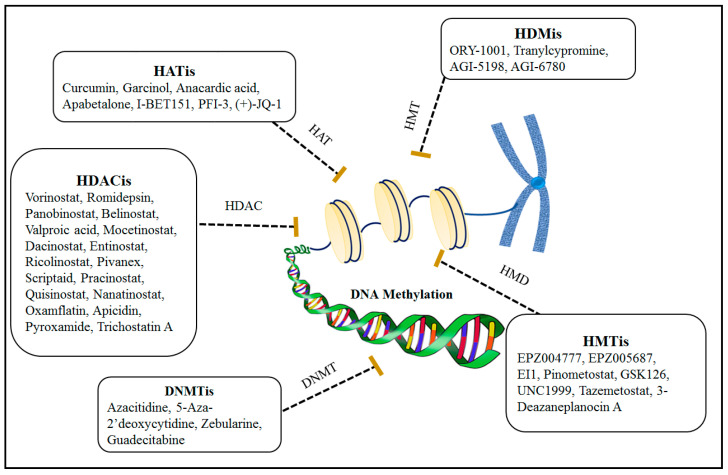
The schematic figure shows the epigenetic drugs used to inhibit the main four classes of enzymes in epigenetic therapy: DNMTi, HMTis, HDMis, HATi, and HDACi.

**Figure 5 genes-14-00873-f005:**
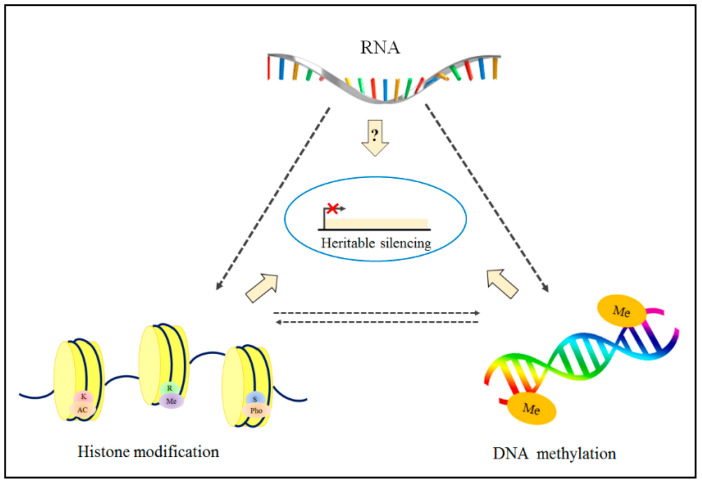
The connection between RNA, histone changes, and the heritable quieting of DNA methylation. Histone deacetylation and other changes in histones, specifically the methylation of lysine 9 inside the amino acid chain of histone (H3-K9), take place in the tails of the histone, causing chromatin compaction and suppressing transcription. Histone modification might interact with DNA methyl-transferases to establish cytosine methylation, which increases the number of histone modifications that prefer silencing. The yeast and plant experiments clearly demonstrated the contribution of RNA interference in the development of heterochromatic states and silencing. As a result, activation of the genetically silent RNA may play a role in the organisms.

**Table 1 genes-14-00873-t001:** Epigenetic drugs to inhibit DNMT, which contains information about the name of drug/alternate name, formula, and applicable in treatment of various diseases, and whether they are approved by the FDA or not.

Categories	Epi-Drug Name	Alternate Name	Conditions	Formula	FDA Approved	Ref.
DNMTi	Azacitidine	5-azacitidine	AML, CML and MDS	$C_{8}H_{12}N_{4}O_{5}$	Yes	[120,137]
	5-Aza-2′-deoxycytidine	Decitabine	AML, CML and MDS	$C_{8}H_{12}N_{4}O_{4}$	Yes	[123]
	Zebularine	NSC309132	Cancers	$C_{9}H_{12}N_{2}O_{5}$	No	[127]
	Decitabine	ASTX727	AML, CML and MDS	$C_{8}H_{12}N_{4}O_{4}$	No	[133]
	Epigallocatechin gallate	EGCG	Cancers	$C_{22}H_{18}O_{11}$	No	[136]
	Guadecitabine	SGI-110	Cancers	$C_{18}H_{24}N_{9}O_{10}P$	No	[128,129]

**Table 2 genes-14-00873-t002:** Epigenetic drugs to inhibit into four groups of enzymes (HMTi, HDMi, HATi and HDACi).

Categories	Epi-Drug Name	Alternate Name	Conditions	Formula	FDA Approved	Ref.
HMTi	EPZ004777	-----------	MLL-translocated leukemia	$C_{28}H_{41}N_{7}O_{4}$	No	[138]
	EPZ015938	Pemrametostat	Cancers, Hematologic malignancies	$C_{24}H_{32}N_{6}O_{3}$	No	[141]
	BIX-01294	-----------	Cancers	$C_{28}H_{38}N_{6}O_{2}$	No	[146]
	UNC0638	-----------	Cancers, anti-viral	$C_{30}H_{47}N_{5}O_{2}$	No	[146]
	BRD4770	-----------	Cancers	$C_{25}H_{23}N_{3}O_{3}$	No	[144]
	EPZ005687	-----------	MM	C_32_H_37_N_5_O_3_	No	[142]
	EI1	KB-145943	Large and follicular B-cell lymphomas, Cancers	$C_{23}H_{26}N_{4}O_{2}$	No	[189]
	Pinometostat	EPZ5676	AML, ALL, MDS	$C_{30}H_{42}N_{8}O_{3}$	No	[139]
	GSK126	GSK2816126A	MM	$C_{31}H_{38}N_{6}O_{2}$	No	[142]
	UNC1999	-----------	Cancers, Hematologic malignancies	C_33_H_43_N_7_O_2_	No	[190,191]
	Tazemetostat	EPZ-6438	Advance epithelioid sarcoma	$C_{34}H_{44}N_{4}O_{4}$	Yes	[147]
	3-Deazaneplanocin A	DZNEP	Cancers	$C_{12}H_{14}N_{4}O_{3}$	No	[143]
HDMi	ORY-1001	-----------	Cancers and AML	$C_{15}H_{24}{\mathrm{Cl}}_{2}N_{2}$	No	[192]
	Tranylcypromine *	Dl-Tranylcypromine	Depression and endometriosis	C_9_H_12_NO_2_S_0.5_	No *	[193,194]
	Enasidenib	AG-221	AML	$C_{19}H_{17}F_{6}N_{7}O$	Yes	[149]
	AGI-5198	IDH-C35	Cancers	$C_{27}H_{31}{\mathrm{FN}}_{4}O_{2}$	No	[195]
	AGI-6780	-----------	AML and cancers	$C_{21}H_{18}F_{3}N_{3}O_{3}S_{2}$	No	[196]
HATi	Curcumin	Diferuloylmethane	Cancers, MM	$C_{21}H_{20}O_{6}$	No	[152]
	Garcinol	Garcinia gummi-gutta fruit	Cancers	$C_{38}H_{50}O_{6}$	No	[112]
	Anacardic acid	Hydroginkgolic acid	Cancers	$C_{22}H_{36}O_{3}$	No	[112]
	Apabetalone	RVX-208	Diabetes Atherosclerosis	$C_{20}H_{22}N_{2}O_{5}$	No	[156]
	I-BET151	GSK1210151A	Cancers, MM	$C_{23}H_{21}N_{5}O_{3}$	No	[197,198]
	PFI-3	-----------	Cancers and lymphoma	C_19_H_19_N_3_O_2_	No	[199]
	(+)-JQ-1	JQ1	Cancers	$C_{23}H_{25}{\mathrm{ClN}}_{4}O_{2}S$	No	[154]
HDACi	Vorinostat	SAHA	CTCL	$C_{14}H_{20}N_{2}O_{3}$	Yes	[115]
	Romidepsin	Depsipeptide	CTCL and PTCL	$C_{24}H_{36}N_{4}O_{6}S_{2}$	Yes	[166,167]
	Panobinostat	LBH589	MM	$C_{21}H_{23}N_{3}O_{2}$	Yes	[163]
	Belinostat	PXD-101	PTCL	$C_{15}H_{14}N_{2}O_{4}S$	Yes	[162]
	Valproic acid **	Sodium valproate	Seizures, cancers	$C_{8}H_{16}O_{2}$	No	[168]
	Mocetinostat	MGCD0103	Cancers, MDS	$C_{23}H_{20}N_{6}O$	No	[200]
	Dacinostat	LAQ824	Cancers and AML	$C_{22}H_{25}N_{3}O_{3}$	No	[201,202]
	Entinostat	MS-275	Cancers, Hematologic malignancies	$C_{21}H_{20}N_{4}O_{3}$	No	[203]
	Ricolinostat	ACY-1215	Cancers and MM	$C_{24}H_{27}N_{5}O_{3}$	No	[204]
	Pivanex	AN-9	Cancers	$C_{10}H_{18}O_{4}$	No	[205]
	Scriptaid	GCK 1026	Cancers and TBI	$C_{18}H_{18}N_{2}O_{4}$	No	[206,207]
	Pracinostat	SB939	Cancers, Hematologic malignancies	$C_{20}H_{30}N_{4}O_{2}$	No	[208]
	givinostat	ITF-2357	Cancers	$C_{24}H_{27}N_{3}O_{4}$	No	[165]
	Resminostat	RAS2410	CTCL	$C_{16}H_{19}N_{3}O_{4}S$	No	[165]
	abexinostat	PCI-24781	Cancers, Hematologic malignancies	$C_{21}H_{23}N_{3}O_{5}$	No	[165]
	MPT0E028	-----------	B-cell lymphomas, Cancers	$C_{17}H_{16}N_{2}O_{4}S$	No	[209]
	Quisinostat	JNJ-26481585	Cancers, Hematologic malignancies	$C_{21}H_{26}N_{6}O_{2}$	No	[210]
	Nanatinostat	CHR-3996	Cancers	$C_{20}H_{19}{\mathrm{FN}}_{6}O_{2}$	No	[211]
	Oxamflatin	Metacept-3	Cancers	$C_{17}H_{14}N_{2}O_{4}S$	No	[212]
	Apicidin	OSI 2040	Cancers	$C_{34}H_{49}N_{5}O_{6}$	No	[213]
	Pyroxamide	-----------	Cancers, Hematologic malignancies	$C_{13}H_{19}N_{3}O_{3}$	No	[214]
	Trichostatin A	TSA	Cancers	$C_{17}H_{22}N_{2}O_{3}$	No	[215]

* Tranylcypromine has been FDA approved as an antidepressant drug. ** Valproic acid has been FDA approved as an antiseizure drug.

## Data Availability

Not applicable.

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
