# Peer review of "An Update of Epigenetic Drugs for the Treatment of Cancers and Brain Diseases: A Comprehensive Review"

_genes, 2023, doi:10.3390/genes14040873_

Round 1

Reviewer 1 Report

The manuscript entitled “Progress of epigenetic drugs for the treatment of cancers and neurodegenerative diseases: A comprehensive review” by Mohammadi Z. S. et al. set out to review information/literature on epigenetics and diseases that are caused by epigenetic factors. The review is structured under the following major sections; 1. Introduction, 2. Epigenetic Therapies, 3. Discussion, and 4. Conclusion, and the Abstract is structured as Introduction, Methods, Results, and Conclusions. The authors concluded that numerous studies have linked epigenetics with various diseases, and that designing appropriate drugs could provide novel strategies for the management of these diseases in near future. The manuscript will need a major overhaul .

Comments

1. The authorship of the manuscript has the same name being repeated for the first and second authors.

2. The manuscript is not a systematic review, so I do not understand the basis for the Abstract was structured into Introduction, Methods, Results, and Discussion. Indeed, the Methods section of the Abstract reads “The purpose of this review article is to provide comprehensive information about the different types of diseases caused by alterations in epigenetic factors such as DNA methylation and histones acetylation/methylation.”, which is obviously not a methodology. The title however suggests that the manuscript/review is on “Progress of epigenetic drugs for the treatment of cancers and neurodegenerative diseases: A comprehensive review”. The two information are obviously not the same.

3. The authors in the Results section of the Abstract that “Consequently, the knowledge to modify aberrant DNA methylation as well as aberrant histone acetylation or methylation via inhibitors by using epigenetic drugs can be a suitable therapeutic approach for a number of diseases.”. Reviewing the literature, as has been done in this manuscript, will not provide new knowledge on epigenetic modifications, so I do not understand how that could be results.

4.  In my opinion, the way the information has been organized in the manuscript, it largely fits a book chapter than a review article; two long paragraphs of epigenetic overview and history, and an Introduction section of about 17 pages.

5. “Epigenetic modification on chromatin” and “Applications of epigenetics in personalized medicine” are appearing two times (with the same information) on pages 4 and another one which is labelled page 2 of 39. The numbering of the document is problematic.

6. The discussion section is merely a summary of the literature that has been reviewed; nothing was really discussed.

7. The manuscript is grammatically not well-written, and poorly organized as well, to say the least. The manuscript needs editing by a native English-speaking person.

8. The authors do not know how to provide headings/titles to Figures and Tables, and they need to seek help with that.

9. Figure 4 is shown but no reference/mention has been made in the text of the manuscript.

Author Response

Reviewer #1

The manuscript entitled “Progress of epigenetic drugs for the treatment of cancers and neurodegenerative diseases: A comprehensive review” by Mohammadi Z. S. et al. set out to review information/literature on epigenetics and diseases that are caused by epigenetic factors. The review is structured under the following major sections; 1. Introduction, 2. Epigenetic Therapies, 3. Discussion, and 4. Conclusion, and the Abstract is structured as Introduction, Methods, Results, and Conclusions. The authors concluded that numerous studies have linked epigenetics with various diseases, and that designing appropriate drugs could provide novel strategies for the management of these diseases in near future. The manuscript will need a major overhaul .
First of all, we thank the reviewer for the valuable comments. We have revised the manuscript according to the reviewer’s helpful comment.

Comments

  1. The authorship of the manuscript has the same name being repeated for the first and second authors.

We are sorry for the mistake. This has been corrected.

  1. The manuscript is not a systematic review, so I do not understand the basis for the Abstract was structured into Introduction, Methods, Results, and Discussion. Indeed, the Methods section of the Abstract reads “The purpose of this review article is to provide comprehensive information about the different types of diseases caused by alterations in epigenetic factors such as DNA methylation and histones acetylation/methylation.”, which is obviously not a methodology. The title however suggests that the manuscript/review is on “Progress of epigenetic drugs for the treatment of cancers and neurodegenerative diseases: A comprehensive review”. The two information are obviously not the same.

You are completely right. However, some journals require the abstract to be structured as “introduction, methodology”. We made changes in response to your insightful feedback. 

  1. The authors in the Results section of the Abstract that “Consequently, the knowledge to modify aberrant DNA methylation as well as aberrant histone acetylation or methylation via inhibitors by using epigenetic drugs can be a suitable therapeutic approach for a number of diseases.”. Reviewing the literature, as has been done in this manuscript, will not provide new knowledge on epigenetic modifications, so I do not understand how that could be results.

This abstract was structured. It has been modified based on your comment. Introduction, methodology, results, and conclusion have been removed from the abstract.

  1. In my opinion, the way the information has been organized in the manuscript, it largely fits a book chapter than a review article; two long paragraphs of epigenetic overview and history, and an Introduction section of about 17 pages.

In fact, a full review covers the whole subject, and naturally the file size increases. The introduction part has been sub-sectioned.

  1. “Epigenetic modification on chromatin” and “Applications of epigenetics in personalized medicine” are appearing two times (with the same information) on pages 4 and another one which is labelled page 2 of 39. The numbering of the document is problematic.

We are really sorry for the mistake. This part has been copied twice from the original file to the journal’s template. Thanks a lot for your correction.  

Regarding the numbering, yes, you are right. The problem is caused by landscape figure. This will be corrected by the journal’s editorial office.

  1. The discussion section is merely a summary of the literature that has been reviewed; nothing was really discussed.

In the research paper, the discussion comprises comparing the obtained results with others and mentioning the strengths and importance of the work compared to others.
But this is a review article, and the important points of past research related to the topic of the article were mentioned in the discussion, and the author cannot create any result.

  1. The manuscript is grammatically not well-written, and poorly organized as well, to say the least. The manuscript needs editing by a native English-speaking person.

Thanks for your comment.  The manuscript has been checked for grammar and improved by a native English speaker.

  1. The authors do not know how to provide headings/titles to Figures and Tables, and they need to seek help with that.

The heading for tables and legend for figures have been revised. Some of the excess description has been removed. If you have any feedback, I would appreciate it. Thanks.

  1. Figure 4 is shown but no reference/mention has been made in the text of the manuscript.

Thank you; figure 4 (now Fig. 3) has been appropriately referenced in the text. 

Reviewer 2 Report

This review article by Mohammadi et al discusses the various types of epigenetic mechanisms that control gene expression, how they have been implicated in disease mechanisms (primarily cancer and neurodegenerative disorders), and the current landscape of small molecule inhibitors available to interfere with aberrant epigenetic processes. While the topic is interesting and a nice compilation of various methods to block the activity of various chromatin remodeling proteins, the article needs substantial revisions and reorganization to improve readability and relevance of content.

1.     The author list includes the first author twice and leaves out an author described in the author contributions list – Shahin Ramazi.

2.     The title does not describe well the content. “Progress of” could be replaced with “An update” and, while the article discusses some neurological disorders linked to epigenetics, none of the drugs described were addressed for their potential use in neurological disorders. All were discussed in the context of cancer.

3.     Section 2 (Epigenetic therapy) and Section 3 (Discussion) were written in a much better style than Section 1. The transitions between topics were better and the content was better organized. The authors that contributed most to Sections 2 and 3 should put some time into editing Section 1. The Introduction was poorly organized and seemed to be listing facts without a sense of telling a story or continuity in content. For instance, a discussion of CpG islands and DNA methylation is interrupted by a discussion of personalized medicine, before going back to discussions of the DNMT gene family.

4.     The subsections “Epigenetic modifications on chromatin” and “Application of epigenetics in personalized medicine” are included twice – lines 162-265 and again in lines 270-375.

5.     Citation of Venkatesan et al is repeated twice (numbers 82 and 83)

6.     Generally, the discussion of non-coding RNAs and RNA modifications does not seem necessary for this review, since therapeutics in this area are under-developed or not available. This reviewer advises removing those topics, focusing on histone modifications and DNA methylation, and developing the content for those more substantially. The actual discussion of the function of different histone modifying proteins is under-developed and lacking details that would make it easier for the reader to understand the targets of the drugs that are later discussed.

7.     Figure 2 is not necessary

8.     Figure 4 is not referenced in the text

9.     Figure 5A has the (A) subpanel label shifted to the middle of the image and 5B is difficult to read due to small print

10.  Abbreviations should be in parentheses. For example, “Runt-related transcription factor RUNX” should be “Runt-related transcription factor (RUNX)”

11.  The personalized medicine subsection (line 220) is very disjointed. The content doesn’t seem to naturally transition from one topic to another and is confusing to read.

12.  The discussion of various syndromes are sometimes only weakly linked to epigenetics. Clearer relationships should be made. The gene implicated in Coffine Lowry syndrome (RSK2) is not identified and neither is its mechanism that relates it to epigenetics. Similarly, the syndromes described are all neurological or multi-system in nature, but they are not later discussed with regards to therapeutic treatment with epigenetic modifying drugs. On the other hand, the discussion of drugs focuses on cancer, but the role of aberrant epigenetic regulation in cancer is not directly discussed in the manuscript, but only mentioned periodically. This leaves the manuscript feeling unfocused.

13.  There is excessive use of subtitles in section 2. Each drug doesn’t need its only subtitle, especially if content is only 1-2 sentences each. A stronger introduction of the HMTs, HATs, HDACs etc processes either as an introduction to the drug category or earlier in the manuscript would better help the reader understand the mechanism(s) of action for the drugs. For instance, a G9a inhibitor is discussed in the section about HMTis, but there was never a clear discussion of why G9a is important in epigenetics and histone modifications.

14.  In Table 1, please verify the chemical formulas for the drugs. The formula for Zebularine is listed as being the same as azacytidine, but it should be C9H12N2O5 EGCG also appears to be incorrect. Similarly, please validate details in Table 2

15.  Overall, it seems as if translation software may have been used, resulting in inappropriate word choices. There are also some typographical errors and grammatical errors too. Below is a list of examples. This is not a list of all problem words, but some of the more obvious ones:

Line 45: “fetus of fruit flies” should be “larvae of fruit flies”

Line 48: “differential legislation of gene function” should be “differential regulation of gene function”

Line 165 is an incomplete sentence and lacks a subject

Line 237: “invasion of life-threatening diseases” should be “evasion of life-threatening diseases”

Line 387: “reappearance-free and generally good endurance in HCC patients” should be “recurrence-free survival in HCC patients”

Line 396: “penetration of cancer cells” should be “invasion of cancer cells”

Line 527: “might be bound” should be “might be found”

Line 528: “histone post-modification” should be “histone post-translational modification”

Line 530: “after-death-mind tissues” should be “post-mortem brain tissues”

Line 565: “maniacal patients” should be “patients with mania.” People are not defined by their disease, but they are people with a disease

Line 580 is a sentence fragment

Line 606: “merged to create multiracial structures” should be “merged to create multiradial structures”

Line 629: “X-linked” not “X-associated”

Line 636-637: “huge mental hindrance and engine brokenness” should be “significant mental disability and motor impairment.” People do not have engines

Line 664: “Crane facial” should be “craniofacial”

Line 670: “cohesin perplexing” should be “cohesin complex”

Line 727 – 728: “Mutations have been identified in the gene that include joint site mutations, minor deletions or insertions of incomprehensible mutations, and false mutations.” This should be “Mutations have been identified in the gene that include splice site mutation, minor deletions or insertions causing frameshift mutations, and nonsense mutations.”

Line 742: The word “turbulence” has no relevant use in this sentence

Line 755: “placed in close proximity to” should be “exposed to” or “added to”

Line 766: MyeloDysplastic Syndrome (MDS) is not a lymphoma. It is a pre-leukemic state. The cells impacted are myeloid in origin, not lymphoid.

Line 906: “deacetylation” not “distillation”

Author Response

Reviewer #2

This review article by Mohammadi et al discusses the various types of epigenetic mechanisms that control gene expression, how they have been implicated in disease mechanisms (primarily cancer and neurodegenerative disorders), and the current landscape of small molecule inhibitors available to interfere with aberrant epigenetic processes. While the topic is interesting and a nice compilation of various methods to block the activity of various chromatin remodeling proteins, the article needs substantial revisions and reorganization to improve readability and relevance of content.

Answer: First of all, we thank the reviewer for the valuable comments. We have revised the manuscript according to the reviewer’s helpful comment.

  1. The author list includes the first author twice and leaves out an author described in the author contributions list – Shahin Ramazi.

 We are sorry for the mistake. This has been corrected.

  1. The title does not describe well the content. “Progress of” could be replaced with “An update” and, while the article discusses some neurological disorders linked to epigenetics, none of the drugs described were addressed for their potential use in neurological disorders. All were discussed in the context of cancer.

Regarding the replacement of “an update” instead of ‘progress of ‘, it has been revised.

Almost every epigenetic-based drug (whether FDA approved or not) has been developed for cancer. It could be because cancer is a fatal disease that can strike at any age. 

  1. Section 2 (Epigenetic therapy) and Section 3 (Discussion) were written in a much better style than Section 1. The transitions between topics were better and the content was better organized. The authors that contributed most to Sections 2 and 3 should put some time into editing Section 1. The Introduction was poorly organized and seemed to be listing facts without a sense of telling a story or continuity in content. For instance, a discussion of CpG islands and DNA methylation is interrupted by a discussion of personalized medicine, before going back to discussions of the DNMT gene family.

Thanks for your valuable comment. The introduction part has been reorganized.  “Application of epigenetics in personalized medicine” is now at the end of the introduction part.

  1. The subsections “Epigenetic modifications on chromatin” and “Application of epigenetics in personalized medicine” are included twice – lines 162-265 and again in lines 270-375.

We are really sorry for the mistake. This part has been copied twice from the original file to the journal’s template. Thanks a lot for your correction. 

  1. Citation of Venkatesan et al is repeated twice (numbers 82 and 83)

Sorry for the mistake. It has been corrected. All the references have been revised and updated.

  1. Generally, the discussion of non-coding RNAs and RNA modifications does not seem necessary for this review, since therapeutics in this area are under-developed or not available. This reviewer advises removing those topics, focusing on histone modifications and DNA methylation, and developing the content for those more substantially. The actual discussion of the function of different histone modifying proteins is under-developed and lacking details that would make it easier for the reader to understand the targets of the drugs that are later discussed.

Although the drugs against aberrant non-coding RNA are still under development (as you mentioned), , they play a vital role in the regulation of epigenetic modification. They have been discussed much less than the other subjects.

  1. Figure 2 is not necessary

It has been removed, following by your suggestion.

  1. Figure 4 is not referenced in the text.

Thank you; figure 4 (now Fig. 3) has been appropriately referenced in the text. 

  1. Figure 5A has the (A) subpanel label shifted to the middle of the image and 5B is difficult to read due to small print

This figure (now referred to as Fig. 4) has been redrawn. 

  1. Abbreviations should be in parentheses. For example, “Runt-related transcription factor RUNX” should be “Runt-related transcription factor (RUNX)”.

The authors are grateful for pointing out this matter. We edited and double-checked the entire text. 

  1. The personalized medicine subsection (line 220) is very disjointed. The content doesn’t seem to naturally transition from one topic to another and is confusing to read.

The authors are grateful for pointing out this matter. We have edited and improved related texts.

  1. The discussion of various syndromes are sometimes only weakly linked to epigenetics. Clearer relationships should be made. The gene implicated in Coffine Lowry syndrome (RSK2) is not identified and neither is its mechanism that relates it to epigenetics. Similarly, the syndromes described are all neurological or multi-system in nature, but they are not later discussed with regards to therapeutic treatment with epigenetic modifying drugs. On the other hand, the discussion of drugs focuses on cancer, but the role of aberrant epigenetic regulation in cancer is not directly discussed in the manuscript, but only mentioned periodically. This leaves the manuscript feeling unfocused.

Thanks for your valuable comment. The epigenetic mechanism of Coffine-Lowry syndrome and ATR-X has been revised.

  1. There is excessive use of subtitles in section 2. Each drug doesn’t need its only subtitle, especially if content is only 1-2 sentences each. A stronger introduction of the HMTs, HATs, HDACs etc processes either as an introduction to the drug category or earlier in the manuscript would better help the reader understand the mechanism(s) of action for the drugs. For instance, a G9a inhibitor is discussed in the section about HMTis, but there was never a clear discussion of why G9a is important in epigenetics and histone modifications.

Epigenetic drugs have been categorized into five groups: DNMTi (A), HMTis (B), HDMis (C), HATi (D), and HDACi (E). DNMTi drugs are designated by the letters A1, A2

G9a, a histone methyltransferase responsible for histone H3 lysine 9 (H3K9) mono- and demethylation. The inhibitors of BRD4770 and BIX-01294 as drug to inhibit G9a were discussed in lines 830-834 (new version) as HMTis inhibitors (group B), however, the function of G9a as a histone methyl transferease was discussed in histone modification.

  1. In Table 1, please verify the chemical formulas for the drugs. The formula for Zebularine is listed as being the same as azacytidine, but it should be C9H12N2O5EGCG also appears to be incorrect. Similarly, please validate details in Table 2

We are sorry for the mistake. All items in the table have been rechecked and approved

  1. Overall, it seems as if translation software may have been used, resulting in inappropriate word choices. There are also some typographical errors and grammatical errors too. Below is a list of examples. This is not a list of all problem words, but some of the more obvious ones:

Thanks a lot for your correction. They have been implemented in the text.

Line 45: “fetus of fruit flies” should be “larvae of fruit flies”

Line 48: “differential legislation of gene function” should be “differential regulation of gene function”

Line 165 is an incomplete sentence and lacks a subject

Line 237: “invasion of life-threatening diseases” should be “evasion of life-threatening diseases”

Line 387: “reappearance-free and generally good endurance in HCC patients” should be “recurrence-free survival in HCC patients”

Line 396: “penetration of cancer cells” should be “invasion of cancer cells”

Line 527: “might be bound” should be “might be found”

Line 528: “histone post-modification” should be “histone post-translational modification”

Line 530: “after-death-mind tissues” should be “post-mortem brain tissues”

Line 565: “maniacal patients” should be “patients with mania.” People are not defined by their disease, but they are people with a disease

Line 580 is a sentence fragment

Line 606: “merged to create multiracial structures” should be “merged to create multiradial structures”

Line 629: “X-linked” not “X-associated”

Line 636-637: “huge mental hindrance and engine brokenness” should be “significant mental disability and motor impairment.” People do not have engines

Line 664: “Crane facial” should be “craniofacial”

Line 670: “cohesin perplexing” should be “cohesin complex”

Line 727 – 728: “Mutations have been identified in the gene that include joint site mutations, minor deletions or insertions of incomprehensible mutations, and false mutations.” This should be “Mutations have been identified in the gene that include splice site mutation, minor deletions or insertions causing frameshift mutations, and nonsense mutations.”

Line 742: The word “turbulence” has no relevant use in this sentence

Line 755: “placed in close proximity to” should be “exposed to” or “added to”

Line 766: MyeloDysplastic Syndrome (MDS) is not a lymphoma. It is a pre-leukemic state. The cells impacted are myeloid in origin, not lymphoid.

Line 906: “deacetylation” not “distillation”

Reviewer 3 Report

Given the recent advances of epigenetics and the development of therapeutics targeting this area of genetic regulation, this review detailing diseases associated with deregulated epigenetics and the use of epigenetic modifiers is timely. The review is well written in some areas and easy to read; however, other areas are unclear and contain typos and grammatical errors. A selection of these are highlighted below, but there are other areas of awkward sentence structure that should be addressed. 

Comments:

1.     Introduction, line 45, not sure of meaning of “history” and epigenetics is not a “phrase”, but a word as you point out in the first sentence of the introduction. 

2.     Legend of Figure 1 is comprehensive but reads more like a paragraph in the introduction and does not adequately describe specific elements in the graphic.

3.     The cellular/pathological model used for concluding m6A was found to be involved in the processing of miRNAs in ref. 15 should be noted in lines 110-112. This way, the reference of “a similar process” related to cancer can be further understood to what the Authors are comparing this study to. 

4.     Incorrect use of “effective” in line 159, suggest, “has a highly influential role” or “environment greatly impacts human health”

5.     The Authors begin to transition from histone modifications and variants to discuss epigenetic profiles within a person and among persons and how this feature can be leveraged for therapeutic development. However, it is confusing to introduce CpG methylation which was discussed in an earlier section. Suggest condensing these five sentences down and using it to introduce the next section and/or using histone variants as the transition. Histone variants appear to be tissue-specific. There is a recent review on histone variants the Authors should consider citing or reviewing to include additional primary literature (Talbert PB and Henikoff S, J Cell Sci, 2021, PMCID: PMC8015243). 

6.     While Figure 3 is a nice summary graphic, it appears to have been enlarged without maintaining aspect ratio and is now stretched out, which makes it difficult to read the text.

7.     Formatting notes

a.     Most of the sub-sections appear to be all under the main section of Introduction, recommend numbering or making main sub-sections to help differentiate which sections are a part of the Introduction and which are a part of the main points of the review. A couple of the sub-sections are formatting differently with a number or as bolded font instead of the italics used throughout.

b.     There appears to be a duplication of two pages in between figures 3 and 4 from previous sections (pages 8 and 9 of the PDF). And no reference to Fig. 4 in the text. 

c.     D-1 and D-2 are a part of the same subsections as written on line 854, “D-1, D-2”, suggest making them one subsection or separate subsections.

d.     In Table 2, some of the drug names are bolded and others are not, and the * and ** endnotes are not referenced next to the drugs in the table, please add in (e.g. Valproic acid**). 

8.     Sentences in lines 388 – 397 regarding review of DNMT1 expression and deregulation in pancreatic tissues and cancer need to be re-organized for clarity.

9.     Figure 5B is difficult to read and the legend needs to be corrected for grammar/repetition. 

10.  The section ATR-X Syndrome starting on line 652 is unclear and not focused on the epigenetic link other than brief mention of MeCP2 and “annihilate the stability of the nucleosome”, please revise. Similar to some of the following sections, Cornelia de Lange CdL Syndrome and Coffine Lowry Syndrome. There seems to be more description of the diseases rather than an emphasis on how epigenetics are linked to the disease and directly correspond to the phenotypes seen. 

11.  Use of phrase “placed in close proximity” on lines 755-756 is incorrect, cancer cells are treated with certain doses…similarly “placed near” on line 856

12.  CTA is defined twice, lines 759-760

13.  Clarification on how Decitabine degrades RB for “destroying cancer cells” and the relevance to epigenetic mechanisms of action is needed (lines 779-780).

14.  For the description of the HMTis, might be more clear and helpful if the connection to the effects on the chromatin and hence changes in epigenetic regulation are mentioned, otherwise, this appears to be a reiteration of the Table rather than a discussion on this drugs.

15.  Select typos and grammatical errors:

a.     Abstract: 

                                               i.     lines 13-14; “include” should be “including” or alternatively, can leave “include” and add in “,which” play crucial roles

                                              ii.     line 14; “researches” does not need to be plural, recommend: Numerous research has been carried out….

                                             iii.     line 16,  recommend “However” be replaced with something along the lines of “These studies indicate”

                                            iv.     line 22, recommend “human cancer’s evolution” be replaced with “the evolution of human cancer”, grammatical error as cancer does not “own” evolution as would be indicated by the apostrophe “s”. 

                                             v.     line 24, abbreviations should be placed within parenthesis, (DNMTs), (HATs), etc. Note, this occurred throughout the manuscript. 

                                            vi.     line 31, “synergy effects” should be “synergistic effects” 

b.     Introduction:

                                               i.     Line 55, here, “Epigenetic”, should be “Epigenetics”; this occurs a couple more times throughout the manuscript, line 85 for instance. 

                                              ii.     Lines 58, usage of i.e. is incorrect, should be e.g. or use the phrase “for example” as in line 56. 

                                             iii.     Line 88, “M6A”, should be “m6A” and again on line 100. 

                                            iv.     Line 134 sub-header typo, “2.”; remove the “.”

                                             v.     Sentence in lines 144-147 is awkward and a run-on. 

                                            vi.     Line 150, unsure what is meant by “as a good disorder”, do Authors mean, when these mechanisms are deregulated 

                                           vii.     Line 163, the “5” is missing the prime character

                                          viii.     Sentence starting on line 165 is a fragment, please revise

                                            ix.     Sentence starting on line 186 is unclear with the “affecting some of these PTMS…”, perhaps delete affecting and change has to have

                                             x.     Sentence starting on line 204 is unclear. Perhaps Authors meant “modifications” rather than “variations” to make a distinction for then going on to discuss histone variants

                                            xi.     Line 504, recommend replacing “that make” with “indicating” and insert “is”: “indicating LSD1 is a putative oncogene”

                                           xii.     Sentences starting on line 526 and 625 are unclear, please revise

                                          xiii.     Line 796, Burkett’s lymphoma should be Burkitt lymphoma

Author Response

Reviewer #3

Given the recent advances of epigenetics and the development of therapeutics targeting this area of genetic regulation, this review detailing disease associated with deregulated epigenetics and the use of epigenetic modifiers is timely. The review is well written in some areas and easy to read; however, other areas are unclear and contain typos and grammatical errors. A selection of these are highlighted below, but there are other areas of awkward sentence structure that should be addressed. 

 Answer: First of all, we thank the reviewer for the valuable comments. We have revised the manuscript according to the reviewer’s helpful comment.

Comments:

  1. Introduction, line 45, not sure of meaning of “history” and epigenetics is not a “phrase”, but a word as you point out in the first sentence of the introduction. 

Thanks for your comment. The “word” has been removed. That sentence gives a brief history of epigenetics, first described in 1940 by Prof. C. Waddington

  1. Legend of Figure 1 is comprehensive but reads more like a paragraph in the introduction and does not adequately describe specific elements in the graphic.

Thanks for your nice and valuable comment. The legend of figure 1 is completely rewritten and summarized.

  1. The cellular/pathological model used for concluding m6A was found to be involved in the processing of miRNAs in ref. 15 should be noted in lines 110-112. This way, the reference of “a similar process” related to cancer can be further understood to what the Authors are comparing this study to. 

In the paper by R. Schickel et al Oncogene (2008) 27, 5959–5974. (now ref 12). The author mentioned miRNAs, which are small non-coding RNAs that are involved in various cell functions, both vital (such as differentiation) and pathological (such as tumorigenesis). 

  1. Incorrect use of “effective” in line 159, suggest, “has a highly influential role” or “environment greatly impacts human health”

This sentence goes along with Fig 2. has been removed following the instruction of another reviewer.

  1. The Authors begin to transition from histone modifications and variants to discuss epigenetic profiles within a person and among persons and how this feature can be leveraged for therapeutic development. However, it is confusing to introduce CpG methylation which was discussed in an earlier section. Suggest condensing these five sentences down and using it to introduce the next section and/or using histone variants as the transition. Histone variants appear to be tissue-specific. There is a recent review on histone variants the Authors should consider citing or reviewing to include additional primary literature (Talbert PB and Henikoff S, J Cell Sci, 2021, PMCID: PMC8015243). 

Thanks for your comments. We used this literature to improve our review article.  (Ref 27)

  1. While Figure 3 is a nice summary graphic, it appears to have been enlarged without maintaining aspect ratio and is now stretched out, which makes it difficult to read the text.

Thanks for your comments. You are right. This figure (now fig 2) does not have good graphics.  This figure is drawn with high quality, but when copied as a landscape, it does not fit in the article’s format. This figure will be sent in the form of a PowerPoint file along with the revised version of the article as a supplemental.

  1. Formatting notes
  2. Most of the sub-sections appear to be all under the main section of Introduction, recommend numbering or making main sub-sections to help differentiate which sections are a part of the Introduction and which are a part of the main points of the review. A couple of the sub-sections are formatting differently with a number or as bolded font instead of the italics used throughout.

Thanks for the good comment. The sections in the introduction were all numbered. The introduction is also slightly modified.

  1. There appears to be a duplication of two pages in between figures 3 and 4 from previous sections (pages 8 and 9 of the PDF). And no reference to Fig. 4 in the text. 

We are really sorry for the mistake. This part has been copied twice from the original file to the journal’s template. Thanks a lot for your correction. 

  1. D-1 and D-2 are a part of the same subsections as written on line 854, “D-1, D-2”, suggest making them one subsection or separate subsections

Answer:  D1 and D2 correspond to garcinol and anacardic acid, respectively. Because the two targets are similar, they are placed in a sub-section. According to your opinion, it was placed in the D1 group.

  1. In Table 2, some of the drug names are bolded and others are not, and the * and ** endnotes are not referenced next to the drugs in the table, please add in (e.g. Valproic acid**). 

Thank you for your attention. The font of the table has been corrected. Items * were mentioned in the “FDA approved” section of the table, which was moved to the “Epi-Drug name” section.

  1. Sentences in lines 388 – 397 regarding review of DNMT1 expression and deregulation in pancreatic tissues and cancer need to be re-organized for clarity.

Thanks. This correction has been carried out.

  1. Figure 5B is difficult to read and the legend needs to be corrected for grammar/repetition. 

This figure (now is fig 4) has been redrawn, and the legend has been revised. 

  1. The section ATR-X Syndrome starting on line 652 is unclear and not focused on the epigenetic link other than brief mention of MeCP2 and “annihilate the stability of the nucleosome”, please revise. Similar to some of the following sections, Cornelia de Lange CdL Syndrome and Coffine Lowry Syndrome. There seems to be more description of the diseases rather than an emphasis on how epigenetics are linked to the disease and directly correspond to the phenotypes seen. 

Thank you for your good comment. Section ATR-X has been improved. The mechanism of the disease, which is related to the regulation of histone H3 and DNA methylation, has been added.  

The epigenetic mechanism of Coffine Lowry Syndrome has been added to the text.

  1. Use of phrase “placed in close proximity” on lines 755-756 is incorrect, cancer cells are treated with certain doses…similarly “placed near” on line 856

We really appreciate your comment. It has been corrected.  

  1. CTA is defined twice, lines 759-760

Thanks. It has been corrected.  

  1. Clarification on how Decitabine degrades RB for “destroying cancer cells” and the relevance to epigenetic mechanisms of action is needed (lines 779-780).

The epigenetic mechanism of Decitabine has been revised. It forms an irreversible covalent complex with DNA (cytosine-5-)-methyltransferase 1 (DNMT1), which leads to the degradation of the enzyme.    

  1. For the description of the HMTis, might be more clear and helpful if the connection to the effects on the chromatin and hence changes in epigenetic regulation are mentioned, otherwise, this appears to be a reiteration of the Table rather than a discussion on this drugs.

  1. Select typos and grammatical errors:

Thanks a lot for your correction. They have been implemented in the text.

  1. Abstract: 
  2. i.lines 13-14; “include” should be “including” or alternatively, can leave “include” and add in “,which” play crucial roles
  3. ii.line 14; “researches” does not need to be plural, recommend: Numerous research has been carried out….

                                             iii.     line 16,  recommend “However” be replaced with something along the lines of “These studies indicate”

  1. line 22, recommend “human cancer’s evolution” be replaced with “the evolution of human cancer”, grammatical error as cancer does not “own” evolution as would be indicated by the apostrophe “s”. 
  2. line 24, abbreviations should be placed within parenthesis, (DNMTs), (HATs), etc. Note, this occurred throughout the manuscript. 
  3. line 31, “synergy effects” should be “synergistic effects” 
  4. Introduction:
  5. Line 55, here, “Epigenetic”, should be “Epigenetics”; this occurs a couple more times throughout the manuscript, line 85 for instance. 
  6. Lines 58, usage of i.e. is incorrect, should be e.g. or use the phrase “for example” as in line 56. 

                                             iii.     Line 88, “M6A”, should be “m6A” and again on line 100. 

  1. Line 134 sub-header typo, “2.”; remove the “.”
  2. Sentence in lines 144-147 is awkward and a run-on. 
  3. vi.Line 150, unsure what is meant by “as a good disorder”, do Authors mean, when these mechanisms are deregulated 

                                           vii.     Line 163, the “5” is missing the prime character

                                          viii.     Sentence starting on line 165 is a fragment, please revise

  1. Sentence starting on line 186 is unclear with the “affecting some of these PTMS…”, perhaps delete affecting and change has to have
  2. Sentence starting on line 204 is unclear. Perhaps Authors meant “modifications” rather than “variations” to make a distinction for then going on to discuss histone variants
  3. Line 504, recommend replacing “that make” with “indicating” and insert “is”: “indicating LSD1 is a putative oncogene”

                                           xii.     Sentences starting on line 526 and 625 are unclear, please revise

                                          xiii.     Line 796, Burkett’s lymphoma should be Burkitt lymphoma

Round 2

Reviewer 2 Report

There are some modest improvements in this revised submission for a review article on epigenetics, the role of epigenetics in diseases, and epigenetic-modifying compounds. The incorrect information has been corrected and word choices are now more appropriate. However, there are some points of concern that remain.

1.     There are still similar issues throughout the manuscript with regards to continuity and clarity. For example, mentions of cancer stem cells (page 16) are made without clearly making a link to epigenetic similarities and differences between normal and cancer stem cells. EZH2 is discussed, where they say it is “silenced” (page 15) and its H3K27m3 function is a “key player in shutting down an upstream oncogene” but then it go on to discuss it’s pro-oncogenic functions. This is contradictory information that is not resolved. Discussion of DNMTs in cancer frequently changes between discussions of DNMT1 and DNMT3 (page 12 of the pdf), making it difficult for the reader to focus on the topic being discussed. A discussion of H3K4me is interrupted by a random sentence on H3K27me, before going back to H3K4me (page 16).  These are just examples of poor organization in some sections that make it difficult for the reader to follow, and trust, the information. No significant reorganization of the manuscript, as requested in the first review, has been accomplished, especially for section 2.

2. There are changes to (current) Figure 4 and it is not clear which subpanels are to be included and what they are showing. I think part of this confusion is how the authors chose to represent document changes with the track changes function in Word. Regardless, it’s not clear why some of the connections are being shown that seem irrelevant to chromatin modifying epigenetics (like LATS2 with head and neck cancer, and LINE-1 elements).  This figure is not compelling and does not have a strong focus on epigenetic factors. It seems to describe genes that have been deregulated as a consequence of epigenetic mechanisms in cancer, but in reality, that list is quite long and this diagram is insufficient to convey that information. A similar diagram of epigenetic genes deregulated in cancer (i.e.: EZH2, DNMTs, etc) would be more informative for a review of this focus.

Author Response

Reviewer #2

There are some modest improvements in this revised submission for a review article on epigenetics, the role of epigenetics in diseases, and epigenetic-modifying compounds. The incorrect information has been corrected and word choices are now more appropriate. However, there are some points of concern that remain.

We sincerely thank referee #2 for her/his re-review and helpful comments.

  1. There are still similar issues throughout the manuscript with regards to continuity and clarity. For example, mentions of cancer stem cells (page 16) are made without clearly making a link to epigenetic similarities and differences between normal and cancer stem cells. EZH2 is discussed, where they say it is “silenced” (page 15) and its H3K27m3 function is a “key player in shutting down an upstream oncogene” but then it go on to discuss it’s pro-oncogenic functions. This is contradictory information that is not resolved. Discussion of DNMTs in cancer frequently changes between discussions of DNMT1 and DNMT3 (page 12 of the pdf), making it difficult for the reader to focus on the topic being discussed. A discussion of H3K4me is interrupted by a random sentence on H3K27me, before going back to H3K4me (page 16).  These are just examples of poor organization in some sections that make it difficult for the reader to follow, and trust, the information. No significant reorganization of the manuscript, as requested in the first review, has been accomplished, especially for section 2.

There are still similar issues throughout the manuscript with regards to continuity and clarity. For example, mentions of cancer stem cells (page 16) are made without clearly making a link to epigenetic similarities and differences between normal and cancer stem cells. EZH2 is discussed, where they say it is “silenced” (page 15) and its H3K27m3 function is a “key player in shutting down an upstream oncogene” but then it go on to discuss it’s pro-oncogenic functions.

"There is an obvious correlation between high EZH2 expression level and tumor expansion," line 367-387 discusses. We also found that a few years ago that inhibition of EZH2 in cancer cells can lead to inhibition of cancer spreading.

https://www.nature.com/articles/onc2017309

CBX7, as a Polycomb (PcG) protein, plays an important role in the maintenance of heterochromatin and gene silencing PCG 1 and 2 recognize and interact with inhibitory histone modifications such an H3K9me3 and H3K27me3.

 https://www.ncbi.nlm.nih.gov/pmc/articles/PMC4718713/

https://www.ncbi.nlm.nih.gov/pmc/articles/PMC4187721/

Regarding “shutting down an upstream oncogene,” it should be noted that EZH2 alone has an oncogenic effect; however, H3K27Me3-mediated transcriptional repression of tumor suppressor genes such as CDH1 (E-cadherin), RKIP, RAD51, DAB2IP, RUNX3, and CDKN1c can silence the upstream oncogene.

https://www.nature.com/articles/onc2017309

In lines 414-435, it is discussed that the demethylase 1 (LSD1/KDM1A) lysine-specific demethylase 1 is an epigenetic enzyme whose overexpression is correlated with poor prognosis in a variety of cancers.

https://www.nature.com/articles/onc2017267

https://www.ncbi.nlm.nih.gov/pmc/articles/PMC5354800/

Various studies have shown that inhibition of LSD1 or knockdown of its gene could inhibit the tumor’s proliferation. LSD1 inhibition enhances H3K4 methylation and increases the expression of tumor-suppressor genes.

Discussion of DNMTs in cancer frequently changes between discussions of DNMT1 and DNMT3 (page 12 of the pdf), making it difficult for the reader to focus on the topic being discussed. A discussion of H3K4me is interrupted by a random sentence on H3K27me, before going back to H3K4me (page 16).  These are just examples of poor organization in some sections that make it difficult for the reader to follow, and trust, the information. No significant reorganization of the manuscript, as requested in the first review, has been accomplished, especially for section 2.

In section 2, various neurodegenerative diseases caused by epigenetic abnormalities have been discussed. H3Kme was discussed in lines 414-436. It is mentioned in lines 414-416 that mutation in H3K27me 2 and 3 have been observed in medulloblastoma. Then, the role of overexpression of lysine demethylase 5A (KD5A) in reducing H3K4me in the promoter region of some genes has been discussed (lines 423-436), which is responsible for retinoblastoma or aging (in the telomere region). In section 2.4, the correlation between epigenetic and neurodegenerative diseases (2.4.1-2.4.9) has been discussed. 

  1. There are changes to (current) Figure 4 and it is not clear which subpanels are to be included and what they are showing. I think part of this confusion is how the authors chose to represent document changes with the track changes function in Word. Regardless, it’s not clear why some of the connections are being shown that seem irrelevant to chromatin modifying epigenetics (like LATS2 with head and neck cancer, and LINE-1 elements).  This figure is not compelling and does not have a strong focus on epigenetic factors. It seems to describe genes that have been deregulated as a consequence of epigenetic mechanisms in cancer, but in reality, that list is quite long and this diagram is insufficient to convey that information. A similar diagram of epigenetic genes deregulated in cancer (i.e.: EZH2, DNMTs, etc) would be more informative for a review of this focus.

Thank you, considering your comment, we deleted figure 4 and added information for epigenetic genes deregulated in cancer.

Reviewer 3 Report

Authors adequately addressed the comments.

Author Response

Reviewer # 3

Authors adequately addressed the comments.

We sincerely thank the esteemed referee #3 for reviewing our manuscript and also for her/his positive opinion.